# Metabolomes of mitochondrial diseases and inclusion body myositis patients: treatment targets and biomarkers

Jana Buzkova[1], Joni Nikkanen[1] , Sofia Ahola[1], Anna H Hakonen[1], Ksenia Sevastianova[2,3], Topi Hovinen[1], Hannele Yki-Järvinen[2,3], Kirsi H Pietiläinen[4,5], Tuula Lönnqvist[6], Vidya Velagapudi[7], Christopher J Carroll[1,8] & Anu Suomalainen[1,9,10,*]

## Abstract

Mitochondrial disorders (MDs) are inherited multi-organ diseases with variable phenotypes. Inclusion body myositis (IBM), a sporadic inflammatory muscle disease, also shows mitochondrial dysfunction. We investigated whether primary and secondary MDs modify metabolism to reveal pathogenic pathways and biomarkers. We investigated metabolomes of 25 mitochondrial myopathy or ataxias patients, 16 unaffected carriers, six IBM and 15 non-mitochondrial neuromuscular disease (NMD) patients and 30 matched controls. MD and IBM metabolomes clustered separately from controls and NMDs. MDs and IBM showed transsulfuration pathway changes; creatine and niacinamide depletion marked NMDs, IBM and infantile-onset spinocerebellar ataxia (IOSCA). Low blood and muscle arginine was specific for patients with m.3243A>G mutation. A four-metabolite blood multi-biomarker (sorbitol, alanine, myoinositol, cystathionine) distinguished primary MDs from others (76% sensitivity, 95% specificity). Our omics approach identified pathways currently used to treat NMDs and mitochondrial stroke-like episodes and proposes nicotinamide riboside in MDs and IBM, and creatine in IOSCA and IBM as novel treatment targets. The disease-specific metabolic fingerprints are valuable "multi-biomarkers" for diagnosis and promising tools for follow-up of disease progression and treatment effect.

**Keywords** biomarker; inclusion body myositis; metabolomics; mitochondrial diseases

**Subject Categories** Genetics, Gene Therapy & Genetic Disease; Metabolism; Post-translational Modifications, Proteolysis & Proteomics

## Introduction

Mitochondrial disorders (MDs) are the most common group of inherited metabolic diseases, with exceptional clinical variability. Globally, their minimum birth prevalence is 1 in 2000–5000 individuals (Thorburn, 2004; Gorman et al, 2016). The adult forms present most commonly with neurological or muscular symptoms (Suomalainen, 2011), but their diagnosis is challenging, and treatment options are scarce. Furthermore, the molecular mechanisms of tissue specificity and clinical variability in MDs are unknown. Mitochondrial dysfunction is also a characteristic sign of inclusion body myositis (IBM), which is a sporadic inflammatory muscle disease, the most common acquired myopathy in the elderly with a prevalence of 2–4:100,000 in Nordic countries (Lindgren et al, 2017). Whether the respiratory chain deficiency in IBM contributes to the disease progression is unknown.

Recent data from disease models highlight whole-organismal metabolic remodelling in MDs, and importantly, these aberrant pathways are amenable for interventions with metabolically active cofactors, such as NAD$^+$ precursor vitamin B3 (Khan et al, 2014). In mice with adult-onset mitochondrial myopathy, metabolomics analysis identified major remodelling of folate-driven one-carbon (1C) metabolism, metabolite methylation and transsulfuration (Nikkanen et al, 2016; Khan et al, 2017). In these mice and in a

1 Research Programs Unit, Molecular Neurology, Biomedicum-Helsinki, University of Helsinki, Helsinki, Finland
2 Department of Medicine, University of Helsinki and Helsinki University Hospital, Helsinki, Finland
3 Minerva Foundation Institute for Medical Research, Helsinki, Finland
4 Research Programs Unit, Diabetes and Obesity, Obesity Research Unit, University of Helsinki, Helsinki, Finland
5 Abdominal Centre, Endocrinology, Helsinki University Central Hospital and University of Helsinki, Helsinki, Finland
6 Department of Child Neurology, Children's Hospital, University of Helsinki, Helsinki, Finland
7 Metabolomics Unit, Institute for Molecular Medicine Finland FIMM, HiLIFE, University of Helsinki, Helsinki, Finland
8 Genetics Research Centre, Molecular and Clinical Sciences Institute, St. George's University of London, London, UK
9 Department of Neurosciences, Helsinki University Hospital, Helsinki, Finland
10 Neuroscience Centre, Helsinki Institute Life Science, University of Helsinki, Helsinki, Finland
*Corresponding author. Tel: +358 4717 1965; E-mail: anu.wartiovaara@helsinki.fi

mouse model for infantile-onset spinocerebellar ataxia (IOSCA), metabolic remodelling shifted the whole-cellular dNTP pools in the affected tissues, with potential to contribute to the mtDNA instability in these disorders.

We report here disease-specific metabolomic fingerprints present in the blood and muscle of patients with different primary and secondary mitochondrial disorders, with potential treatment targets, biomarkers and pathogenic pathways.

# Results

## Metabolomic analysis of blood reveals disease-specific biomarker profiles

We performed high-throughput targeted semiquantitative analysis of 94 metabolites in blood samples of patients with mtDNA maintenance disorders (IOSCA; mitochondrial recessive ataxia syndrome [MIRAS]; progressive external ophthalmoplegia/mitochondrial myopathy [PEO]), or defect in mitochondrial translation (mitochondrial myopathy, encephalomyopathy, lactic acidosis and stroke-like episodes [MELAS]/maternally inherited diabetes and deafness [MIDD]); as well as of IBM patients, MIRAS carriers and non-mitochondrial neuromuscular disease (NMD) patients (Table 1, Dataset EV1). The patient and control groups were analysed by the partial least squares–discriminant analysis (PLS-DA; Figs 1 and 2). Metabolites with the highest separation power in PLS-DA were ranked by variable importance in projection (VIP) scores (Figs 1 and 2), described below for each disease.

Infantile-onset spinocerebellar ataxia blood metabolome clustered separately from the controls (Fig 1A). The metabolic profile of this epileptic encephalohepatopathy showed a strong component of creatine, bile acid and transsulfuration pathway changes. Low amounts of creatinine (fold change [FC] $-1.6$, $P < 0.001$), the secreted breakdown product of creatine, suggested increased creatine turnover, which was also supported by significantly increased creatine/ creatinine ratio (Fig 3A), despite the increased amount of creatine in the blood (FC $+1.9$, $P = 0.017$). Decreased steady-state kynurenate (FC $-2.2$, $P = 0.003$) and niacinamide (NAM; FC $-2.0$, $P = 0.013$; Fig 1A) pointed to altered $NAD^+$ synthesis pathway and an increase in $NAD^+$ demand. The serine-driven transsulfuration pathway (Nikkanen et al, 2016) imbalance was marked by increased upstream metabolites, serine (FC $+1.3$, $P = 0.013$), glutamate (FC $+2.4$, $P < 0.001$) and cystathionine (FC $+1.9$, $P = 0.003$), but depletion of transsulfuration-dependent taurine (FC $-1.6$, $P = 0.002$) and reduced form of glutathione (FC $-2.2$, $P = 0.063$). Two bile acids, glycocholic acid (GCA; FC $+2.4$, $P = 0.003$) and taurine-conjugated taurochenodeoxy-cholic acid (TCDCA; FC $+2.6$, $P = 0.015$), were increased (Fig 1A). Glutathione depletion indicates decreased potential for antioxidant capacity in IOSCA. Additionally, we analysed a blood sample of an IOSCA child patient (four years of age) who showed high creatine/ creatinine ratio and low taurine and kynurenate (Fig 3B). The significant depletion of kynurenate and the significant depletion of niacinamide are both consistent with depletion of $NAD^+$.

Mitochondrial recessive ataxia syndrome blood metabolome clustered separately from controls. The patients showed significant increase in carbohydrate derivatives, i.e., sorbitol (FC $+6.2$, $P < 0.0001$), glucuronate (FC $+1.4$, $P = 0.014$) and myoinositol (FC

$+1.2$, $P = 0.017$; Figs 1B and 3A). Other changes included increased alanine (FC $+1.4$, $P = 0.002$) and decreased lysine (FC $-1.2$, $P = 0.002$) and carnosine (FC $-1.4$, $P = 0.034$), involved, e.g., in inactivation of methylglyoxal, a product of high sugars. Similar to IOSCA, cystathionine was increased (FC $+1.5$, $P = 0.004$; Fig 1B), whereas other transsulfuration or creatine metabolites were not significantly changed (Fig 3A).

In PEO patients, the blood metabolome clustered separately from controls (Fig 1C). The significantly changed metabolites included elevated cystathionine (FC $+4.1$, $P < 0.001$), phosphoethanolamine (PE; FC $+1.5$, $P = 0.005$), glutamine (FC $+1.2$, $P = 0.002$) and sorbitol (FC $+2.1$, $P = 0.006$; Figs 1C and 3A). Overall, PEO blood showed a wide upregulation of amino acids and purine precursors (xanthine and xanthosine; both FC $+1.5$) as previously reported (Ahola et al, 2016; Nikkanen et al, 2016) and an increase in $NAD^+$ synthesis pathway (kynurenine [FC $+1.3$, $P < 0.001$]; 3-hydroxy-DL-kynurenine [FC $+2.0$, $P = 0.004$]; Fig 1C). Furthermore, unmethylated metabolite precursor of creatine, guanidinoacetic acid (GAA; FC $+1.5$, $P = 0.012$; Fig 1C), was increased, suggesting deficient metabolite methylation.

The MELAS/MIDD blood metabolome clustered separately from the controls (Fig 1D). The results showed remarkably increased carbohydrate derivatives: sorbitol (FC $+11.1$, $P < 0.001$), glucuronate (FC $+2.0$, $P < 0.001$), myoinositol (FC $+1.6$, $P = 0.003$; Fig 3A) and sucrose (FC $+1.5$, $P = 0.035$). Amino acids were in general higher than controls, including alanine (FC $+1.8$, $P < 0.0001$), with an exception of significantly decreased arginine (FC $-1.6$, $P < 0.0001$; Fig 3A), which was specific for MELAS/MIDD in our material. These changes were not explained by diabetes, as they remained MELAS/MIDD-specific even when we compared all patients with increased insulin resistance to normoglycemic patients (Table EV1).

We then asked whether inclusion body myositis, a sporadic inflammatory muscle disease with secondary findings of mitochondrial myopathy in the muscle (respiratory chain-deficient muscle fibres, multiple mtDNA deletions), would share blood metabolic features with primary respiratory chain deficiencies. IBM blood metabolome clustered separately from controls (Fig 2A). The IBM metabolic profile was defined by elevated cystathionine (FC $+2.7$, $P < 0.0001$), dimethylglycine (FC $+2.6$, $P = 0.001$), TCDCA (FC $+4.6$, $P < 0.001$) and citrulline (FC $+1.4$, $P < 0.001$). The alternative $NAD^+$ synthesis pathway was highly upregulated: kynurenine (FC $+1.6$, $P = 0.002$) and its hydroxylated form, 3-hydroxy-DL-kynurenine (FC $+6.2$, $P < 0.001$), were significantly induced; however, niacinamide was reduced (FC $-2.6$, $P = 0.001$; Fig 2A). Furthermore, IBM showed significant increases in nucleotide synthesis precursors (e.g., adenosine, deoxycytidine, cytidine and cytosine; FC $+2.2$, $+1.3$, $+1.5$, $+3.4$, respectively), as well as carbohydrate derivatives: sucrose, myoinositol and glucuronate (FC $+2.8$, $+1.6$ and $+1.5$, respectively; Figs 2A and 3A). Also, creatine/creatinine ratio was significantly increased, suggesting low creatine pools (Fig 3A). Overall, IBM blood metabolome resembled the respiratory chain deficiencies (IBM shared 39% significantly changed metabolites with PEO) rather than NMDs (IBM and NDMs shared 23% of significantly changed metabolites).

To define which changes were specific for mitochondrial diseases and which were general consequences of muscle disease, we analysed a group of heterogeneous non-mitochondrial neuromuscular diseases. The PLS-DA model of NMD patients clustered separately

**Table 1.  Characteristics of mitochondrial and non-mitochondrial neuromuscular disease patients.**

| | IOSCA (N = 5[a]) | MIRAS (N = 9) | PEO (N = 8) | MELAS/MIDD (N = 5) | IBM (N = 6) | NMD (N = 15) |
|---|---|---|---|---|---|---|
| Gender (n) | 2F, 3M | 2F, 7M | 3F, 5M | 2F, 3M | 3F, 3M | 12F, 3M |
| Age of onset (years)[b] | 1–2 | 29.6 (18.0–44.0) | 27.8 (21.0–35.0) | 39.3 (30.0–48.0) | 61.4 (49.0–83.0) | 26.3 (2.0–60.0) |
| Age at sampling (years)[b] | 38.6 (33.0–42.0) | 41.2 (21–52.0) | 50.0 (39.0–57.0) | 54.0 (39.0–68.0) | 71.0 (58.0–85.0) | 49.9 (23.0–77.0) |
| FGF21 (pg/ml)[c,d] | – | 132.5 (51.0–279.8) | 454.0 (222.0–604.3)[†] | 562.0 (188.5 –2569.0)[‡] | 57.0 (34.3–287.8) | 114.0 (24.0–190.0) |
| Inheritance disease gene, amino acid change | AR, *TWNK* p. Y508C | AR, *POLG* p. W748S+E1143G | AD, *TWNK* 13AA dup (TWNK-PEO); AR, *POLG* p. A1105T/N468D (POLG-PEO); Sporadic, mtDNA single deletion (Del-PEO) | Maternal, mtDNA m.3243A>G, tRNA^Leu (UUR) | Sporadic | AD or AR, DMPK, ZNF9, CAPN3, PABPN1, SMN1, TIA1 or unknown |
| Histological findings in skeletal muscle | None | 1–5% COX-/SDH+ fibres | POLG/TWNK-PEO: 5–12% COX-/SDH+; Del-PEO: 30–60% COX-/SDH+ | 5–30% COX-/SDH+ fibres | 1–8% COX-/SDH+ fibres | Dystrophy, hypertrophy, normal respiratory chain |
| MtDNA consequences | MtDNA depletion in brain and liver | MtDNA depletion, small amount of heteroplasmic multiple mtDNA deletions in skeletal muscle | Heteroplasmic multiple mtDNA deletions, or single large mtDNA deletion in skeletal muscle | Heteroplasmy; ~70% of mutant mtDNA in muscle and urine epithelial cells | Multiple mtDNA deletions | None |
| Muscle symptoms | – | –/+ | + | ++ | ++ | ++ |
| Clinical symptoms | Childhood-onset ataxia, neuropathy, athetosis, hearing loss, epilepsy, hepatopathy | Ataxia, neuropathy, epilepsy, psychiatric symptoms, cognitive decline, obesity/ insulin resistance | Mitochondrial myopathy, ptosis, progressive external ophthalmoplegia, exercise intolerance | Mitochondrial myopathy, cardiomyopathy, diabetes mellitus, hearing loss, stroke-like episodes | Distal progressive muscle weakness | |

–, muscle phenotype not present; **+**, mild muscle phenotype; **++**, primary muscle phenotype. AR, autosomal recessive; AD, autosomal dominant; COX-/SDH+, cytochrome C oxidase-negative/succinate dehydrogenase-positive fibres; F, female; M, male; N, number; mtDNA, mitochondrial DNA.
See Dataset EV1 for raw data.
[a]Additional IOSCA child patient (four years of age; Fig 2B). This patient, however, was not included in the overall statistical analysis due to lack of appropriate age- and gender-matched control samples.
[b]Values represent mean with minimal and maximal age.
[c]Values represent median with interquartile range.
[d]Normal value for FGF21 ≤ 331 pg/ml (Lehtonen et al, 2016).
[‡]$P = 0.009$, [†]$P = 0.002$ (nonparametric Kruskal–Wallis test).

from controls (Fig 2B). The NMD metabolome was discriminated by creatine metabolism (creatine FC +2.1, $P < 0.001$; creatinine FC −1.8, $P = 0.005$) with increased creatine/creatinine ratio (Fig 3A), supporting depleted creatine pool. Furthermore, TCDCA (FC +3.6, $P = 0.005$), folic acid (FC +3.0, $P = 0.005$) and glutamate (FC +1.6, $P = 0.012$) were increased and niacinamide (FC −1.7, $P = 0.004$), spermidine (FC −1.9, $P = 0.006$) and histidine (FC −1.2, $P = 0.006$) were reduced (Fig 2B). However, cystathionine and alanine, increased in both primary (IOSCA, MIRAS, PEO and MELAS) and secondary (IBM) MD patients, were not elevated in NMD patients, or in healthy controls or MIRAS carriers (Fig 3A).

The blood metabolome of the heterozygous carriers of the recessive MIRAS allele showed separation from controls (Fig 2C). Similar to MIRAS patients, they had increased glucuronate (FC +1.2, $P = 0.029$), sorbitol (FC +1.6, $P = 0.029$) and myoinositol (FC +1.3, $P = 0.002$; Fig 3A) and low carnosine (FC −1.8, $P < 0.001$). Cystathionine and alanine, the strongest markers of MIRAS, however, were not increased

(Fig 3A). In general, the metabolic profile of MIRAS carriers revealed subtle but significant changes, including dimethylglycine (FC +1.8, $P < 0.0001$), aspartate (FC +1.8, $P < 0.0001$) and cytosine (FC +1.8, $P < 0.001$; Fig 2C). The results suggest that carrier status of one MIRAS allele is not completely neutral for metabolism.

## Methylation cycle and glutathione pathway are affected in muscle of MD patients

In order to compare the blood metabolomic findings with the primarily affected tissue to understand the tissue-specific changes, we performed targeted semiquantitative analysis of 111 metabolites in muscle from patients and control subjects. Mitochondrial recessive ataxia syndrome is primarily a nervous system disorder; however, the patients carry a small amount of multiple mtDNA deletions in their skeletal muscle (Table 1; Hakonen et al, 2008), similar to PEO patients. MIRAS muscle metabolome was separated from controls in

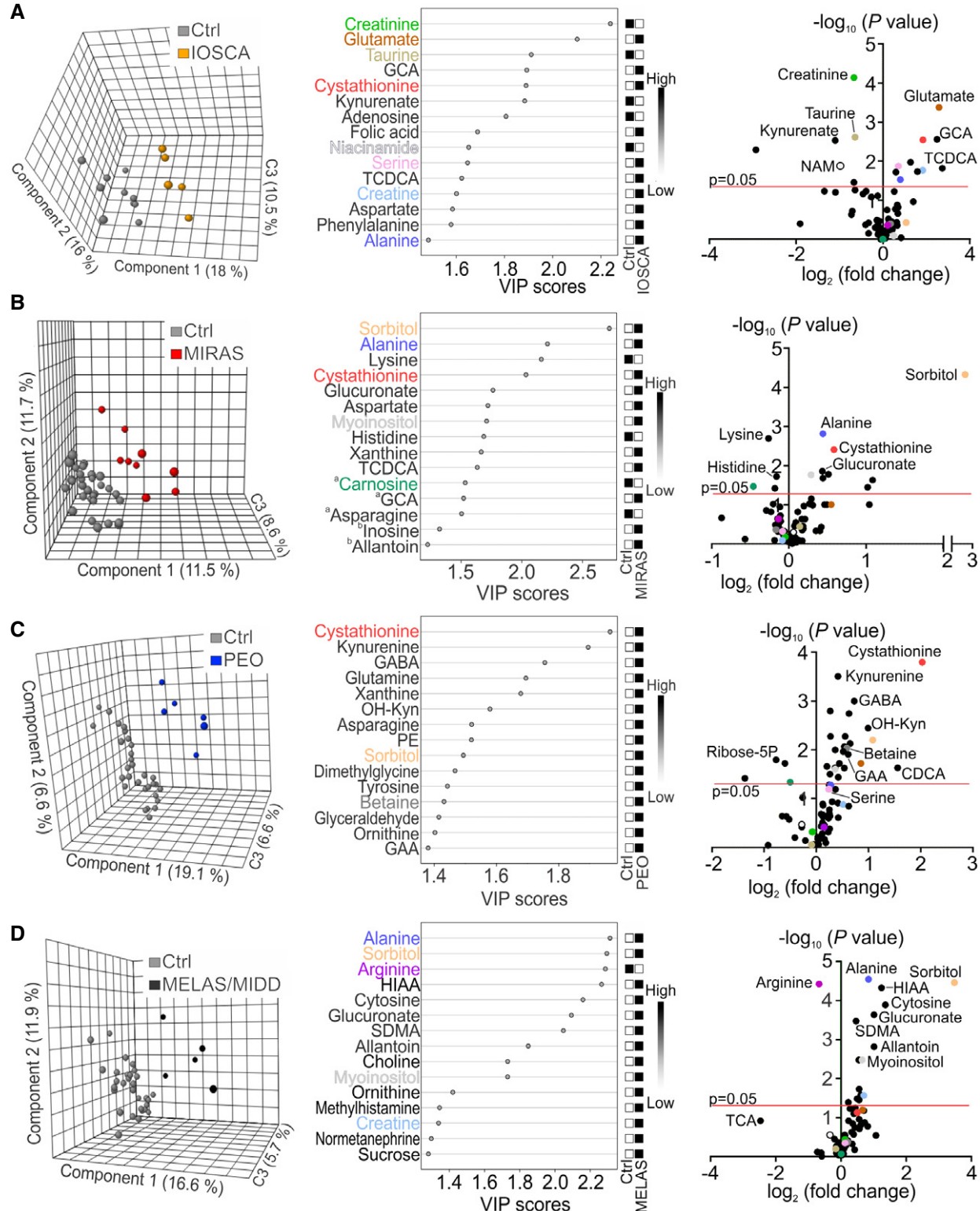

**Figure 1.  Metabolomic fingerprints of primary mitochondrial diseases.**

A–D   Clustering of metabolome data in patients and controls; PLS-DA plots; VIP score plots of top 15 metabolites; volcano plots of all metabolites in blood of IOSCA (A), MIRAS (B), PEO (C), and MELAS/MIDD (D). [a]Significantly changed metabolites outside the FDR cut-off. [b]Metabolites not significantly changed between patients and controls. Colours in VIP score and volcano plots indicate the same most relevant and/or significantly changed metabolites among all patient groups. C3, component 3; CDCA, chenodeoxycholic acid; GABA, γ-aminobutyric acid; HIAA, 5-hydroxyindole-3-acetic acid; OH-Kyn, 3-hydroxy-DL-kynurenine; SDMA, symmetric dimethylarginine; TCA, taurocholic acid. See Dataset EV2 for raw data.

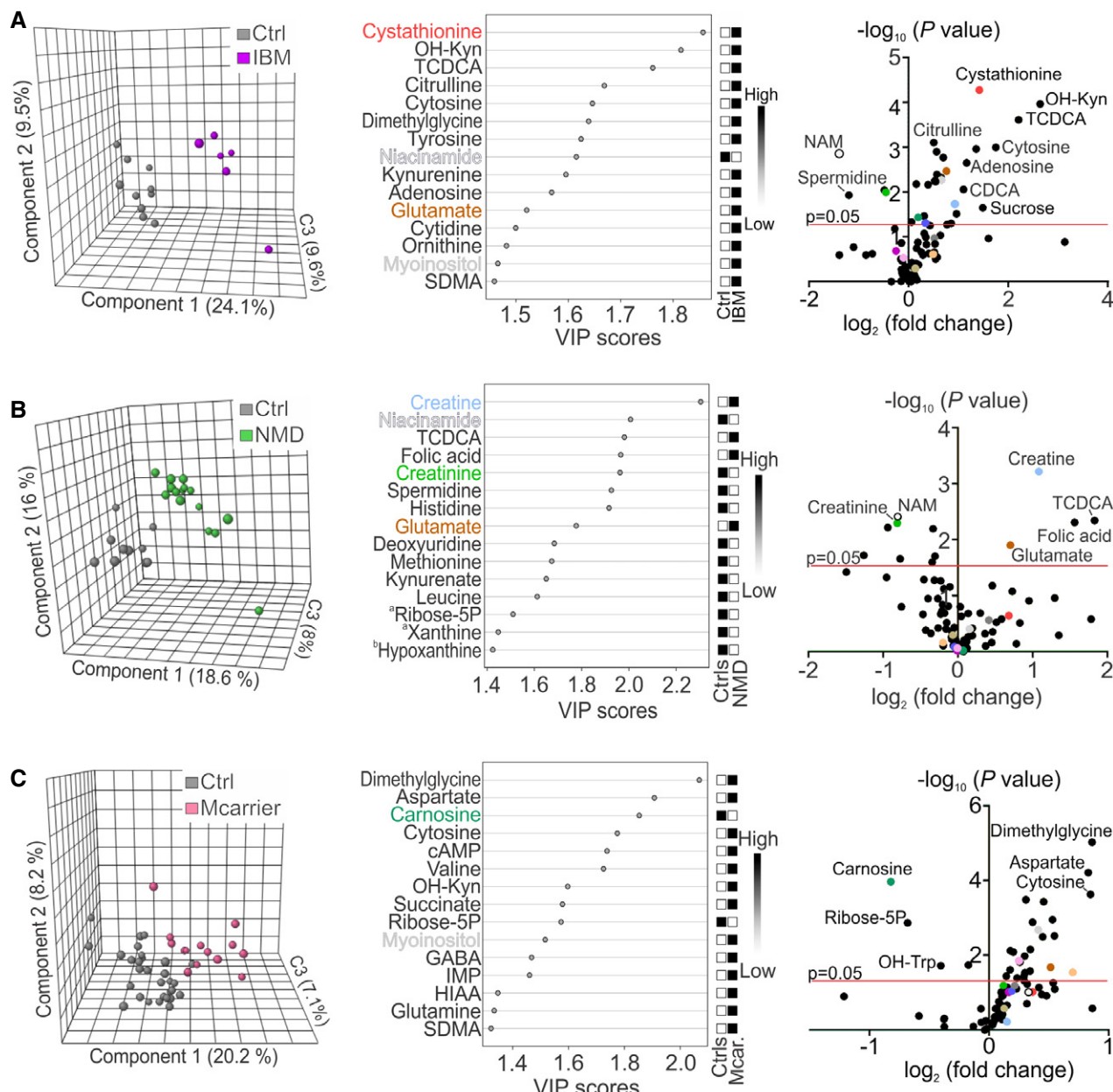

**Figure 2. Metabolomic fingerprints of inclusion body myositis, muscle disorders of non-mitochondrial origin and MIRAS carriers.**

A–C Clustering of metabolome data in patients and controls; PLS-DA plots; VIP score plots of top 15 metabolites; volcano plots of all metabolites in blood of IBM (A), NMD patients (B) and MIRAS carriers (C). [a]Significantly changed metabolites outside the FDR cut-off. [b]Metabolites not significantly changed between patients and controls. Colours in VIP score and volcano plots indicate the same most relevant and/or significantly changed metabolites among all patient groups. cAMP, cyclic AMP; C3, component 3; CDCA, chenodeoxycholic acid; GABA, γ-aminobutyric acid; HIAA, 5-hydroxyindole-3-acetic acid; IMP, inosine monophosphate; OH-Kyn, 3-hydroxy-DL-kynurenine; OH-Trp, hydroxytryptophan; SDMA, symmetric dimethylarginine. See Dataset EV2 for raw data.

PLS-DA (Fig 4A). The most significantly changed metabolites (false discovery rate [FDR] < 0.5) were the major methyl carriers, elevated S-adenosyl-L-homocysteine (SAH; FC +1.8, $P = 0.009$) and reduced S-adenosyl-L-methionine (SAM; FC −3.4, $P = 0.034$; Fig 4A, Dataset EV2). This was an indication for methyl cycle imbalance in MIRAS muscle. However, the MIRAS muscle metabolite signature did not overlap with the blood biomarker profile (Fig 4C), e.g., low carbohydrate derivatives in muscle (Fig 4D), suggesting that the metabolic changes in the blood likely reflected metabolism of another affected

tissue, such as the brain or liver; indeed, muscle manifestation in MIRAS is mild or completely lacking.

The PEO and MELAS/MIDD patients in this study had mainly muscle/cardiac symptoms. The PEO muscle metabolites separated from controls in PLS-DA model (Fig 4B), and the muscle metabolic profile revealed changes in key metabolites of the methyl cycle and glutathione metabolism: cystathionine was remarkably increased (FC +8.3, $P = 0.009$), and methionine (FC +1.5, $P = 0.032$) and serine (FC +1.9, $P = 0.016$; Fig 4B, Dataset EV2) were elevated (FDR < 0.3).

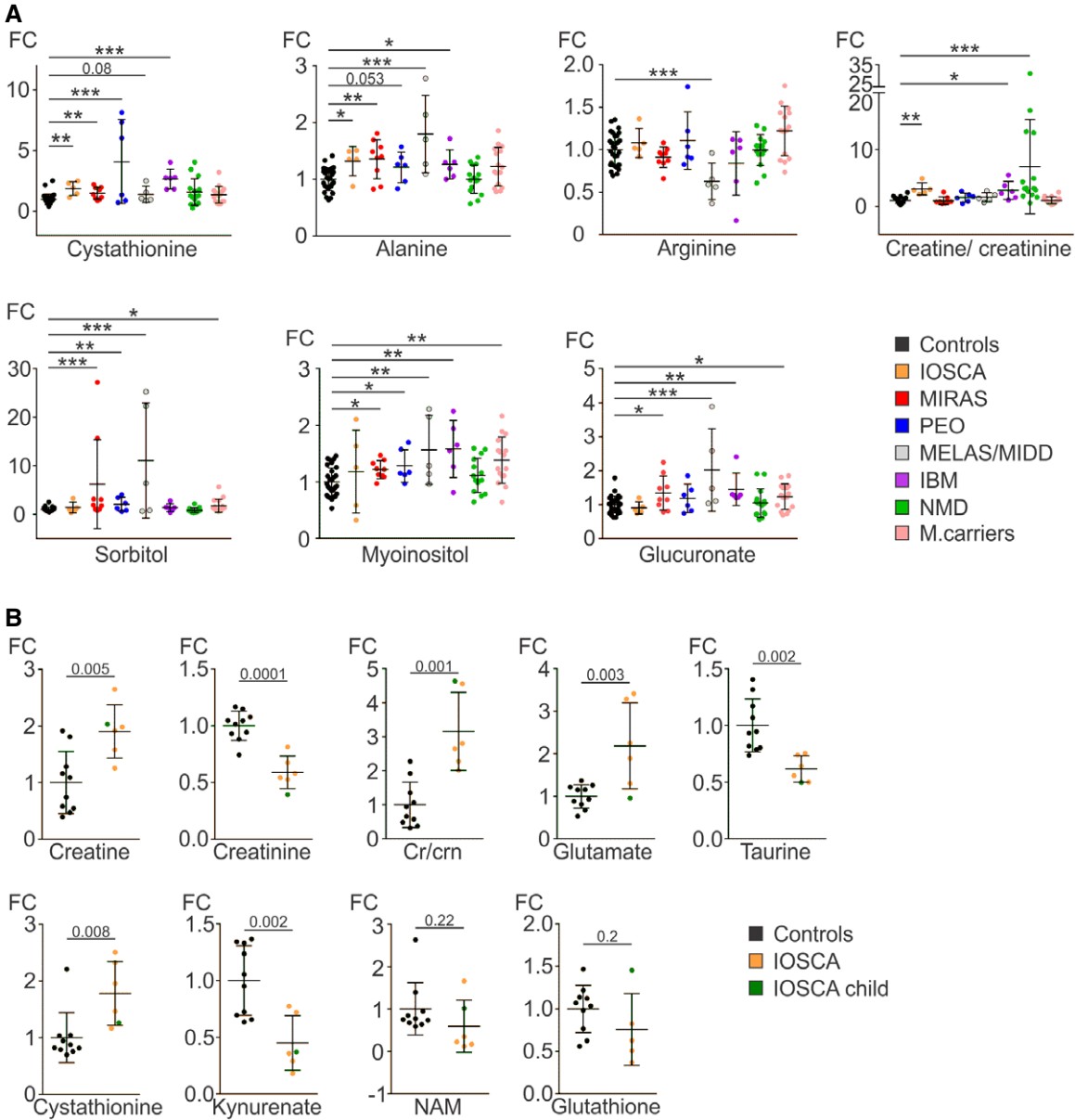

**Figure 3.  Quantification of disease-specific single metabolites in blood.**

A   Relative values of single metabolites and creatine/creatinine ratios in blood of primary MD, IBM and NMD patients, and MIRAS carriers compared to controls.
B   Relative values of single metabolites in blood of adult IOSCA (marked "IOSCA") patients and one IOSCA child patient compared to controls.

Data information: All data represent mean ± SD. For all individual metabolites: *P < 0.05, **P < 0.01, ***P < 0.001 (two-sample t-test). For creatine/creatinine (cr/crn) ratio: *P = 0.022, **P = 0.005, ***P = 0.0001 (Mann–Whitney test). See Dataset EV2 for raw data.

These metabolites overlapped well with the metabolites changed in PEO blood (Fig 4C and D; Nikkanen *et al*, 2016). Similar to PEO, cystathionine was increased in MELAS/MIDD muscle (FC +1.3) as were other contributors to the transsulfuration cycle, namely gamma-glutamyl-cysteine (γ-Glu-Cys; FC +1.4), SAM (FC +1.9) and glutamate (FC +1.2). In contrast, adenosine (FC −3.6), GAA (FC −3.4) and betaine (FC −2.4) were reduced (Dataset EV2). The metabolites from MELAS/MIDD blood and muscle partially overlapped (Fig 4C and D): e.g., low arginine (blood [FC −1.6; Fig 3A] and muscle [FC −2.4; Fig 4D]). The findings of PEO and MELAS/MIDD patients support the

conclusion that the blood metabolome reflects at least partially the metabolome of the disease affected tissue.

## Pathway analysis: transsulfuration metabolism remodelled in mtDNA maintenance disorders

Pathway analysis of the full metabolomes of blood showed several significantly changed pathways common to all MDs, but not to NMDs (Fig 5 shows top 10 pathways with ≥10% metabolites detected in the pathway; Dataset EV3). Transsulfuration pathway

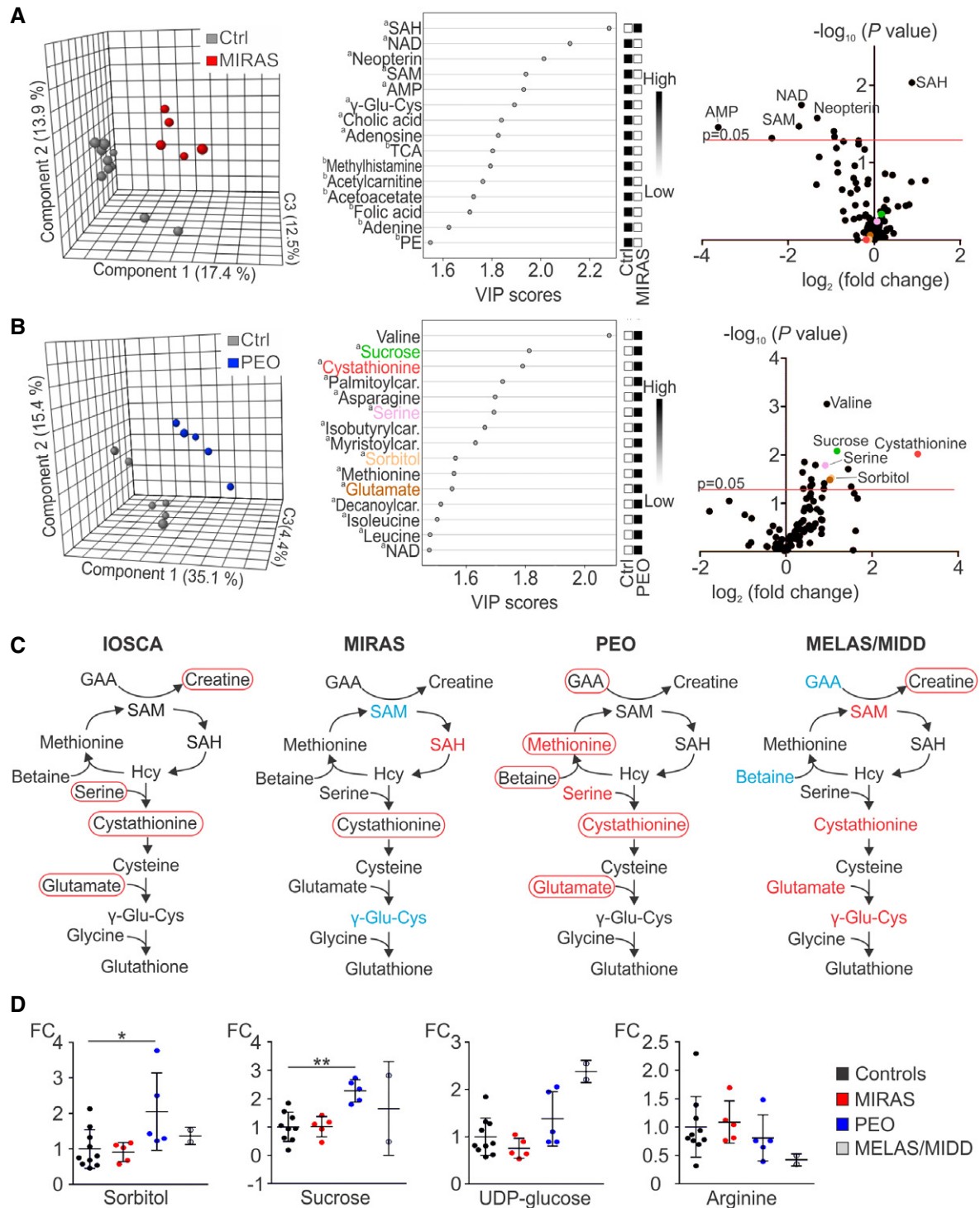

**Figure 4.  Muscle metabolomes of MIRAS, PEO and MELAS patients.**

A, B    Metabolomes of muscle of MIRAS (A) and PEO (B) patients; PLS-DA plots; VIP score plots of top 15 metabolites; volcano plots of all metabolites.

C    Methyl cycle, transsulfuration and glutathione biosynthesis pathways changed in IOSCA, MIRAS, PEO and MELAS/MIDD patients. Circled text: metabolites changed in blood; red, increased; blue, decreased. Coloured text: metabolites changed in muscle; red, increased; blue, decreased. Selected for MELAS muscle were metabolites with the highest fold change.

D    Relative values of metabolites in muscle of MIRAS, PEO and MELAS/MIDD patients compared to controls.

Data information: [a]Significantly changed metabolites outside the FDR cut-off. [b]Metabolites not significantly changed between patients and controls. All data represent mean ± SD. *$P = 0.031$, **$P = 0.008$ (two-sample $t$-test). AMP, adenosine monophosphate; car., carnitine; Hcy, homocysteine; NAD, nicotinamide adenine dinucleotide; TCA, taurocholic acid; UDP, uridine diphosphate. See Dataset EV2 for raw data.

(cysteine and methionine metabolism) and amino acid biosynthesis pathway (alanine, aspartate and glutamate biosynthesis) were aberrant in mtDNA expression disorders (mtDNA maintenance/translation: IOSCA, MIRAS, PEO, MELAS) and IBM (Fig 5A–E), as well as in muscle of PEO patients (Fig 5G), the cysteine and methionine metabolism being among the top four significant pathways in blood, and folate metabolism being also prominent in MIRAS muscle (Fig 5H). Transsulfuration pathway or the amino acid biosynthesis pathway was not significantly changed in NMD patients (Fig 5F). Purine/pyrimidine synthesis was common to muscle-manifesting disorders including NMD (Fig 5C–F; Dataset EV3).

### Sorbitol, myoinositol, alanine and cystathionine: a multi-biomarker for MDs

We then tested the performance of metabolites as disease biomarkers in a pooled set of MD patients (MIRAS, PEO and MELAS/MIDD patients; $n = 20$), asking which metabolites would show the best sensitivity and specificity for MD to distinguish them from healthy controls ($n = 30$). By receiver operating characteristic (ROC) curve analysis, the top four significant metabolites with the highest area under curve (AUC) were sorbitol 0.81 (95% confidence interval [CI] 0.68–0.94, $P = 0.0003$), alanine 0.81 (95% CI: 0.67–0.94, $P = 0.0003$), myoinositol 0.79 (95% CI: 0.66–0.91, $P = 0.0007$) and cystathionine 0.78 (95% CI: 0.65–0.91, $P = 0.001$; Fig 6A), which we together call "multi-biomarker" for MDs. We then compared it to conventional blood biomarkers: lactate and pyruvate, and fibroblast growth factor 21 (FGF21), a serum biomarker of muscle-manifesting MDs. For the same set of patients and controls, FGF21 had the highest AUC 0.87 (95% CI: 0.74–0.99, $P = 0.0001$), followed by lactate 0.86 (95% CI: 0.76–0.97, $P = 0.0001$) and pyruvate 0.78 (95% CI: 0.64–0.93, $P = 0.0017$; Fig 6A). We then compared sensitivity of the four metabolites and the conventional blood biomarkers to identify MDs. FGF21 showed the highest sensitivity of all (68%; 95% CI: 43.5–87.4; Fig 6A), when including all MD patients, and when considering only muscle-manifesting MDs—known to induce FGF21 secretion—its sensitivity in this material was 91% (95% CI: 66.4–100.0; Fig EV1B). Lactate and pyruvate showed sensitivity 45% (95% CI: 23.1–68.5) and 13% (95% CI: 1.6–38.4), respectively, and specificity 97% (95% CI: 82.8–99.9; Fig 6A). As single metabolites, sorbitol and alanine showed sensitivity of 55% (95% CI: 31.5–76.9) and specificity 97% (95% CI: 82.8–99.9), and for myoinositol and cystathionine, the sensitivity was 25% (95% CI: 8.7–49.1), and specificity 93.3% (95% CI: 77.9–99.2) and 97% (95% CI: 82.8–99.9), respectively (Fig 6A), to identify MDs. However, when we combined the four metabolites together and calculated mean centroid values from sorbitol, alanine, cystathionine and myoinositol for all patients and controls, the primary and secondary MDs differed significantly from controls, MIRAS carriers and NMDs (Fig 6B). The sensitivity of this blood multi-biomarker to find primary MDs raised to 76% (95% CI: 54.9–90.6) and specificity to 95% (95% CI: 83.1–99.4) with AUC 0.94 (95% CI: 0.88–0.995, $P = 0.0001$; Fig 6B).

## Discussion

We report here disease-specific metabolomic fingerprints, detectable in blood, of primary mitochondrial muscle and brain disorders,

inclusion body myositis with secondary mitochondrial defects, and a mixed group of severe primary muscle dystrophies/atrophies. Our evidence indicates the following: (i) All the disease groups show blood metabolic fingerprints that cluster separately from healthy controls, indicating the potential of metabolomic fingerprints as multi-biomarkers for diagnosis, follow-up of disease progression and treatment effect; (ii) IBM causes similar global metabolomic changes as primary mitochondrial myopathies reflected in blood, suggesting that metabolic strategies for intervention may be shared in these disease groups; (iii) Heterozygous carriership for the recessive MIRAS allele, common in Western populations (Hakonen *et al*, 2005; Winterthun *et al*, 2005; population frequency 1:84 in Finns and 1:100 in Norwegians; www.sisuproject.fi) is not metabolically neutral; (iv) Our omics approach identified known therapy targets, already in clinical use [arginine in MELAS/MIDD blood and muscle (Koga *et al*, 2005; Koenig *et al*, 2016); creatine in NMDs; Kley *et al*, 2013)] and identified new potential targets for treatment of IOSCA (creatine, glutathione [N-acetyl-cysteine] and NAD$^+$ [nicotinamide riboside] supplementation) and IBM (creatine supplementation), proposing that targeted metabolomic analysis may not only be valuable for mechanistic studies, but also suggest metabolic targets for treatment trials.

The pathogenic mechanism of sporadic IBM, the inflammatory and treatment-resistant muscle disease is still unknown, although it is one of the most frequently encountered muscle diseases in neurology clinics. Typical findings include inflammation, increased number of autophagosomes and characteristics of mitochondrial myopathy: respiratory chain-deficient muscle fibres and accumulation of multiple mtDNA deletions (Oldfors *et al*, 1995). These mitochondrial changes are considered to be a secondary consequence of IBM pathogenesis, probably due to lower turnover of mitochondria as a result of insufficient macroautophagy/mitophagy (Askanas *et al*, 2015), but whether mitochondrial dysfunction in IBM has functional consequences has been unknown. Our finding of the similarity of blood metabolomes of the primary MDs and IBM suggests that mitochondrial dysfunction drives the metabolic changes in IBM reflected in the blood. These findings propose that intervention strategies of mitochondrial biogenesis, NAD$^+$-boosters or rapamycin, suggested to be beneficial for mitochondrial myopathies in mice (Viscomi *et al*, 2011; Yatsuga & Suomalainen, 2012; Cerutti *et al*, 2014; Khan *et al*, 2014, 2017), should be evaluated also in IBM.

A prominent metabolic pattern in different MDs in blood and muscle pointed to aberrant folate-driven 1C-cycle, which is the major cellular anabolic biosynthesis pathway, providing 1C-units for growth and repair. The pathways that feed from this cycle depend on cell-type needs and include *de novo* purine synthesis, methyl cycle, genome and metabolite methylation (creatine and phospholipid synthesis) and transsulfuration (cysteine metabolism, glutathione and taurine synthesis). These 1C-pathways were recently discovered to be remodelled in cells and mice with mtDNA maintenance defects, leading to dNTP pool imbalance, as well as induced glucose-driven *de novo* serine biosynthesis with glucose carbon flux towards glutathione synthesis (Bao *et al*, 2016; Nikkanen *et al*, 2016). The most prominent hits in IOSCA, MIRAS, PEO, MELAS and IBM pointed to aberrant transsulfuration pathway, with the most significant depletion of taurine and reduced form of glutathione found in IOSCA. Related findings were observed also in muscle of MIRAS patients, with more

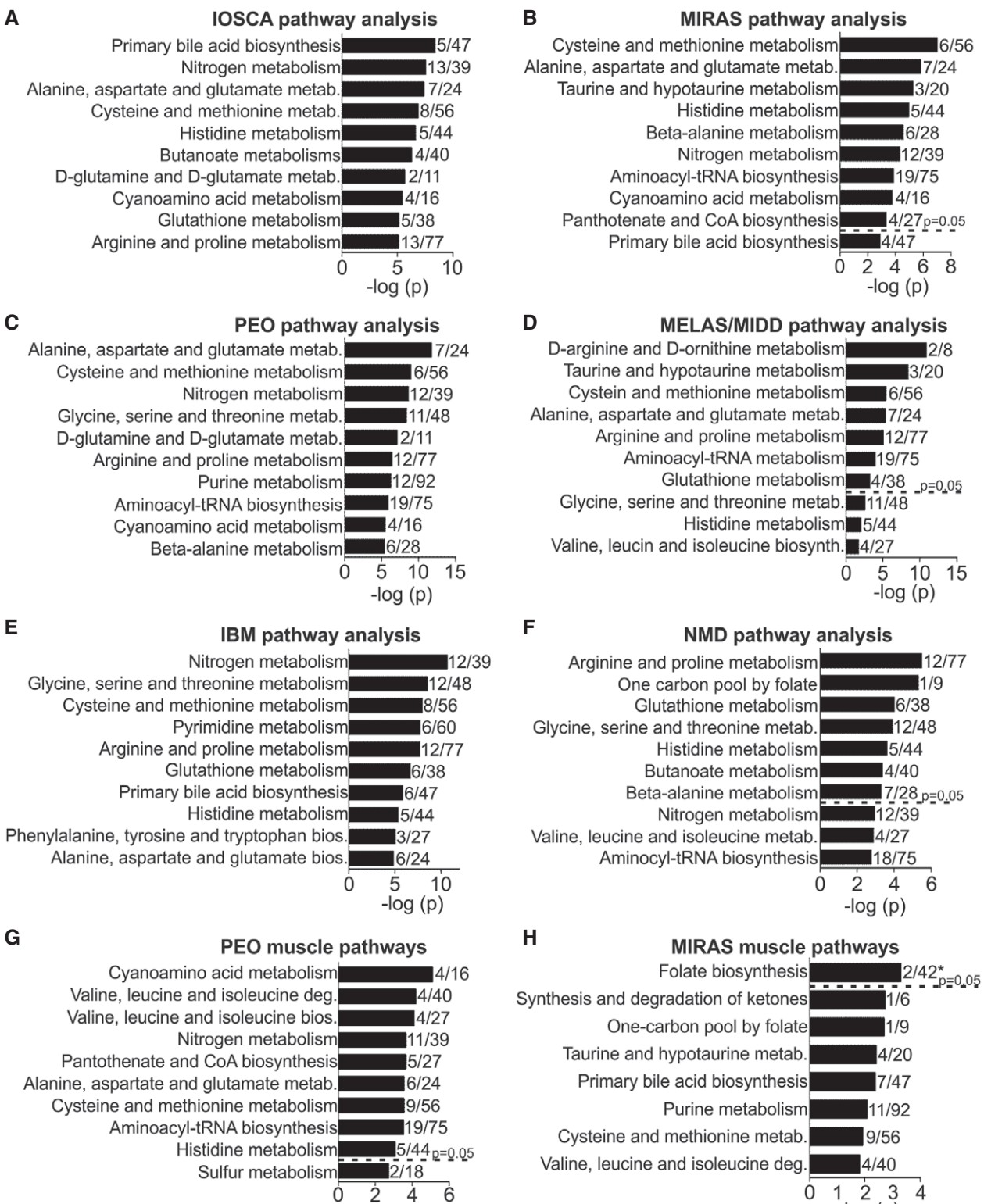

**Figure 5. Pathway analysis of blood and muscle metabolites.**

A–F   Changed metabolic pathways in blood of IOSCA (A), MIRAS (B), PEO (C), MELAS/MIDD (D), IBM (E) and NMD (F) patients.

G, H   Changed metabolic pathways in muscle of PEO (G) and MIRAS (H) patients.

Data information: Top 10 pathways with ≥ 10% of detected metabolites per pathway are shown. *5% metabolite coverage in the pathway. bio., biosynthesis; deg., degradation; metab., metabolism. See Dataset EV3 for raw data.

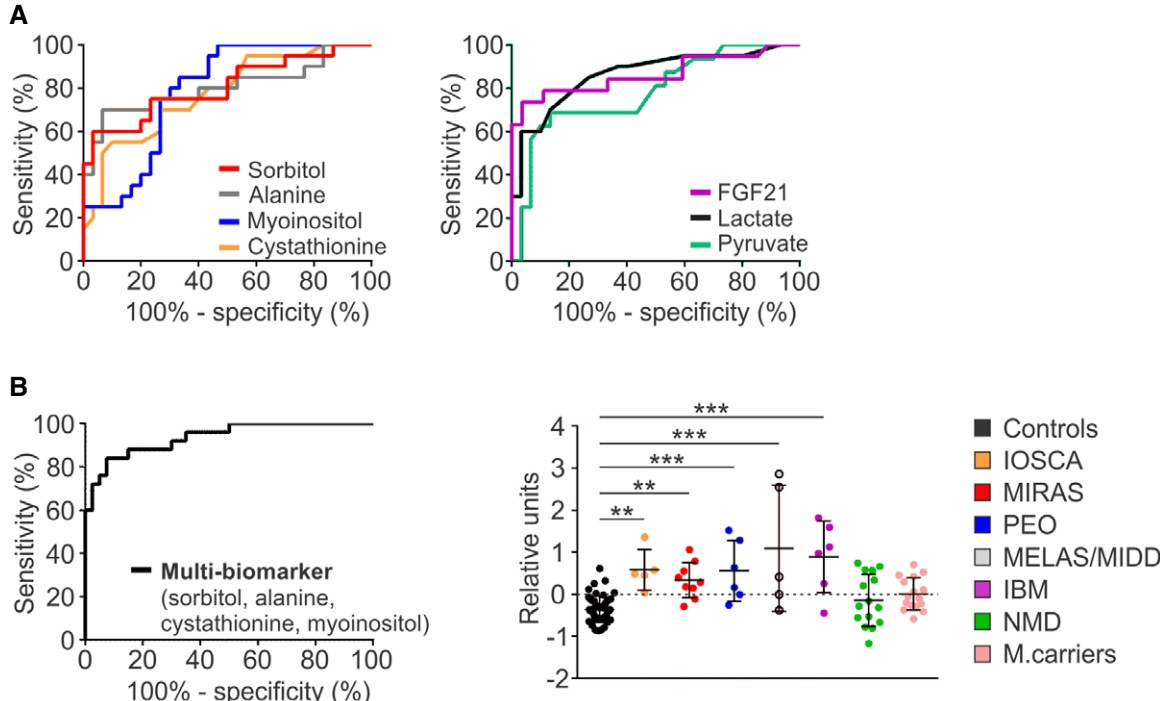

**Figure 6. Blood metabolites as biomarkers for mitochondrial diseases.**

A   ROC curves for individual metabolites sorbitol, alanine, myoinositol and cystathionine (left) and conventional blood biomarkers lactate and pyruvate, and cytokine FGF21 (right) in blood of MIRAS, PEO and MELAS/MIDD patients ($n = 20$) compared to controls ($n = 30$).

B   ROC curve for the combined "multi-biomarker" of sorbitol/alanine/myoinositol/cystathionine for primary MDs compared to controls (left); mean centroids for MD, IBM and NMD patients, and MIRAS carriers compared to controls (right).

Data information: ROC analysis: AUC of sorbitol 0.81 (95% CI: 0.68–0.94, $P = 0.0003$), alanine 0.81 (95% CI: 0.66–0.94, $P = 0.0003$), myoinositol 0.79 (95% CI: 0.66–0.91, $P = 0.0007$) and cystathionine 0.78 (95% CI: 0.65–0.91, $P = 0.001$); AUC of conventional biomarkers: lactate 0.86 (95% CI: 0.76–0.97, $P = 0.0001$) and pyruvate 0.78 (95% CI: 0.64–0.93, $P = 0.0017$), and FGF21 0.87 (95% CI: 0.74–0.99, $P = 0.0001$). AUC of "multi-biomarker" 0.94 (95% CI: 0.88–0.995, $P = 0.0001$). Mean centroid data represent mean ± SD. **$P < 0.01$, ***$P < 0.001$ (1-way ANOVA with Dunnett's multiple comparison test). See Dataset EV2 for raw data.

emphasis in the proximal folate-pool and methyl cycle: low methyl-donor S-adenosyl-methionine and high S-adenosyl-homocysteine point to lowered methylation capacity. Increased levels of carbohydrate-derived metabolites (sorbitol, myoinositol), a sign of high glucose uptake in the muscle, is also known to challenge regeneration of reduced glutathione (Brownlee, 2001). These changes in mtDNA maintenance diseases point to a challenged glutathione supply and suggest that N-acetyl-cysteine supplementation, providing cysteine for glutathione and taurine synthesis, could be tried as a metabolic bypass therapy.

Our unbiased screen identified creatine depletion in NMD patients, which was an interesting proof of principle, as a Cochrane review found creatine supplementation to be useful for muscle dystrophies (Kley et al, 2013). However, similarly low global creatine pool, represented by the blood creatine/creatinine ratio, was found to be present in IBM and also in IOSCA, despite the fact that IOSCA patients do not show any muscle phenotype (Lönnqvist et al, 1998) or low muscle mass (Park et al, 2013). Although muscle inactivity can contribute to increased creatine/creatinine ratio in blood of NMD and IBM patients, the increased creatine/creatinine ratio in adult IOSCA patients, as well as in the IOSCA child patient who is motorically as active as her age-mates, suggests an important role of creatine metabolism in the disease pathogenesis. Creatine synthesis

is a major methyl group user, utilizing the same 1C/methyl pool as transsulfuration cycle, and thus, the creatine supplementation in IOSCA and IBM should be studied.

Our omics approach highlighted a deficiency of arginine to be specific for MELAS/MIDD in both blood and muscle, as the only significantly decreased amino acid. Low arginine has previously been reported in blood of MELAS patients with stroke-like episodes (Koga et al, 2005), but not in MIDD or in patients' tissues. L-arginine supplementation has been reported to prevent and treat MELAS-associated stroke-like episodes in open-label trials (Koga et al, 2005, 2007; Naini et al, 2005; El-Hattab et al, 2012) and was recently recommended as a treatment (Koenig et al, 2016). Arginine acts as a precursor for nitric oxide (NO) that has a major role in muscle relaxation of small blood vessels (Koga et al, 2005), and arginine deficiency and the consequent NO deficiency could contribute to the pathogenesis of stroke-like episodes in MELAS. Our unbiased metabolomics approach supports arginine deficiency to be also a feature of MIDD. This was a second interesting proof of principle of the potential of an omics approach in identifying therapeutically valuable metabolic targets.

We found the full set of ~100 metabolites very informative, and a previous study in mitochondrial myopathy mice supported the biomarker potential of a semiquantitative metabolomic analysis in

follow-up of therapy effect: after treatment with rapamycin, the metabolomes of wild-type and affected mice shifted from separate clusters to overlap (Khan *et al*, 2017). However, we also identified here a minimal set of four individual metabolites that were enough to distinguish MDs from other muscle-manifesting disorders as a "multi-biomarker": cystathionine, sorbitol, myoinositol and alanine. Sorbitol and myoinositol have not been reported previously to be changed in MDs. Elevated cystathionine was found in single patients with mtDNA depletion syndrome (Tadiboyina *et al*, 2005; Mudd *et al*, 2012), but not in blood samples of patients with Leigh syndrome (Thompson Legault *et al*, 2015), caused by a structural defect of the respiratory chain. Alanine is a standard blood biomarker in MDs (Haas *et al*, 2008), but is also found increased in other conditions, including sepsis, tetraspasticity, hyperinsulinism, chronic thiamine deficiency or as a side effect of valproic acid treatment (Thabet *et al*, 2000; Noguera *et al*, 2004; Thauvin-Robinet *et al*, 2004; Morava *et al*, 2006). Despite lacking sensitivity as single metabolites, their power increases as a combined multi-biomarker. We propose their blood values to be tested in follow-up of disease progression and therapy effect when testing of a large-scale targeted metabolome is not feasible.

Increased carbohydrate metabolites, but not cystathionine and/or alanine, were detected in blood of asymptomatic MIRAS carriers. Previously, a cross-sectional screening study of asymptomatic m.3243A>G (MELAS) mutation carriers revealed significant differences in their urinary proteome compared to healthy controls (Hall *et al*, 2015). The evidence suggests that carriership of a recessive nuclear mutation or low mtDNA mutation heteroplasmy level modestly remodels metabolism, and these changes are detectable in blood. Whether these effects have consequences for the health of the carriers (Hakonen *et al*, 2005; Winterthun *et al*, 2005) remains to be studied considering the common occurrence in populations (up to 1:84 for single mutations).

A recent study on muscle metabolomics of a dog model for Duchenne muscular dystrophy reported arginine and proline metabolism as the top changed pathways (Abdullah *et al*, 2017), which is also the top pathway in the blood metabolomes of our NMD patients. These findings suggest that the blood metabolomic responses to muscular dystrophy are conserved in species and that a semiquantitative metabolomics assay—or arginine/proline content of serum—could be useful as a multi-biomarker for treatment follow-up in muscle dystrophies.

A limitation of our study is the small sample size of separate patient groups, which may lead to overfitting in the PLS-DA analysis of metabolome data. However, the groups are genetically homogenous: the patients had a confirmed DNA or morphological diagnosis, and for a rare disease material, our cohort is well representative. In the metabolomic analysis, the patients cluster separately from their age- and gender-matched controls. Furthermore, importantly, the metabolomic data do not stand alone: these human results robustly replicate previous metabolic and proteomic data obtained from different mouse and cell models with related defects, further validated with independent methods in different model systems (Ost *et al*, 2015; Bao *et al*, 2016; Nikkanen *et al*, 2016; Kühl *et al*, 2017). Our results highlight the potential of targeted metabolomics of blood and tissue samples for mechanistic studies and as biomarkers for follow-up of disease progression and treatment effects. Importantly, our omics screen identified targets for metabolite treatment, both verifying previously known targets and suggesting novel ones for IOSCA

and IBM, disorders with few treatment options. Longitudinal follow-up studies to assess metabolome dynamics during disease progression and therapeutic interventions are warranted.

# Materials and Methods

The study was undertaken according to Helsinki Declaration and approved by the ethical review board of Helsinki University Central Hospital (HUCH) with written and signed informed consents from the study subjects.

## Participants

Table 1 summarizes the patient data (Dataset EV1). We obtained plasma samples from nine MIRAS patients (OMIM #607459), and muscle biopsy samples from five of them. All patients were homozygous for the "MIRAS allele" (p.W748S+E1143G) in *POLG*, the nuclear gene encoding the catalytic subunit of the mitochondrial DNA polymerase gamma. MIRAS is an autosomal recessive disorder affecting mainly the central nervous system (CNS). The MIRAS patients in this study manifested typically with progressive gait disturbance, polyneuropathy, ataxia, and some with epilepsy, but signs of muscle pathology were absent or mild (respiratory chain-deficient muscle fibres, mtDNA deletions and blood FGF21 concentration; Table 1; Hakonen *et al*, 2005; Lehtonen *et al*, 2016). We also collected plasma from 16 non-manifesting MIRAS family members heterozygous for the MIRAS allele ("MIRAS carriers", Dataset EV1). The MELAS (OMIM #540000)/MIDD (maternally inherited diabetes and deafness; OMIM #520000) patients carried a heteroplasmic m.3243A>G point mutation in mtDNA tRNA$^{Leu(UUR)}$ gene (Goto *et al*, 1990). Plasma samples were obtained from five MELAS patients and muscle samples from two patients. The patients manifested in the late adulthood (~40 years of age) with different combinations of mitochondrial myopathy and ragged-red fibres (RRFs), cardiomyopathy, diabetes mellitus, hearing loss and stroke-like episodes; showed a high amount of respiratory chain-deficient fibres in their muscle, were heteroplasmic for the mutant mtDNA in the skeletal muscle (range 50–90%) and urine epithelial cells (65–80%) as determined by minisequencing (Suomalainen *et al*, 1993) and showed high FGF21 concentration in their blood (Table 1; the patients were described in Lehtonen *et al*, 2016). Additionally, we utilized six serum samples from patients with inclusion body myositis (IBM; OMIM #147421). IBM is typically a sporadic muscle disease characterized by progressive weakness and wasting of distal muscles, the muscle sample showing inflammation and typical findings of mitochondrial myopathy—a high amount of respiratory chain-deficient muscle fibres—but normal level of blood FGF21 (Table 1; Suomalainen *et al*, 2011; Lehtonen *et al*, 2016). We therefore consider IBM a secondary mitochondrial disease. As "non-mitochondrial disease controls", we analysed serum metabolomes from 15 patients with different neuromuscular diseases (NMDs; Suomalainen *et al*, 2011; Lehtonen *et al*, 2016): Becker's muscle dystrophy (*DMD*), myotonic dystrophy type I (*DMPK*) and II (*ZNF9*), motoneuron disease (unknown), muscle weakness (*CAPN3*), oculopharyngeal muscular dystrophy (*PABPN1*), late-onset Pompe's disease (*GAA*), spinal muscular atrophy type II

**The paper explained**

**Problem**

Mitochondrial disorders are rare, diagnosis challenging and pathophysiology poorly known. Studies in mice with mitochondrial myopathy suggested major systemic metabolomic changes, but human metabolomic studies in genetically and clinically uniform patient groups are unavailable. Furthermore, no knowledge exists of metabolic changes in inclusion body myositis (IBM), a common sporadic muscle disease with secondary mitochondrial myopathy findings. Lastly, sensitive and specific blood biomarkers are lacking.

**Results**

We investigated a representative group of mitochondrial myopathy and ataxia patients, unaffected MIRAS carriers, as well as patients with IBM and non-mitochondrial neuromuscular disease by metabolomic analysis. We identified distinct disease-group-specific metabolomic fingerprints in blood and muscle. IBM clustered together with mitochondrial myopathies, proposing important contribution of mitochondrial dysfunction in IBM-related muscle weakness. A novel four-metabolite multi-biomarker (sorbitol, alanine, cystathionine and myoinositol) distinguished primary and secondary mitochondrial disorders from other groups.

**Impact**

Our omics data highlight the potential of metabolomic fingerprints in blood as multi-biomarkers for diagnosis, disease progression and treatment effect.

(*SMN1*) and III (unknown), and Welander's muscular dystrophy (*TIA1*; Table 1). To compare the parallel disease-specific signatures of all available genetically defined mitochondrial disease groups, we also re-analyse metabolomic data from blood and muscle of patients with progressive external ophthalmoplegia (PEO) and infantile-onset spinocerebellar ataxia (IOSCA; Nikkanen *et al*, 2016). The PEO cohort included patients with autosomal dominant PEO with *TWNK* mutations (TWNK-PEO, OMIM #609286; Spelbrink *et al*, 2001), patients with recessive mutations in *POLG* (POLG-PEO, OMIM #157640; Luoma *et al*, 2004) and patients with a sporadic single heteroplasmic large mtDNA deletion (Del-PEO; Table 1). These patients had typical PEO and mitochondrial myopathy as their uniform clinical phenotype, and our previous studies have indicated PEO patients to respond remarkably similarly to treatments (Ahola *et al*, 2016), strongly suggesting similarity in underlying pathophysiological changes. Therefore, we grouped the PEO patients as a single study group, despite their different genotypes. Muscle samples were obtained from three TWNK-PEO patients and two Del-PEO patients (Dataset EV1). IOSCA (OMIM #271245) is caused by a homozygous recessive mutation in *TWNK* (Nikali *et al*, 2005). Additionally, we were able to obtain a plasma sample from an IOSCA child patient (age 4 years). Plasma samples were obtained from 30 healthy volunteers (mean age 42 years, 22–63 years [min–max age]), serum samples from 10 healthy volunteers (mean age 41 years, 29–57 years [min-max age]) and muscle samples from 10 healthy volunteers (mean age 44 years, 23–55 years [min–max age]).

**Blood and muscle samples**

Blood samples were taken after an overnight fast during an outpatient visit at Helsinki University Hospital. Serum (no coagulant included) and plasma (with K2-EDTA) were immediately separated from the peripheral venous blood by centrifugation at 3,000 *g* at +4°C for 15 min and stored at −80°C until analysis. Muscle samples were taken by needle biopsy from *vastus lateralis* muscle under local anaesthesia, snap frozen and stored at −80°C until analysis.

**Targeted metabolomics analysis**

Serum/plasma and muscle metabolites were extracted and analysed as previously described (Khan *et al*, 2014; Nikkanen *et al*, 2016; Kolho *et al*, 2017; Nandania *et al*, 2018). Briefly, metabolites were extracted from frozen muscle samples (10–35 mg) homogenized with extraction solvent (1:30, sample: solvent) and 100 μl of serum/plasma samples (1:4, sample: solvent), separated with Waters ACQUITY Ultra-Performance Liquid Chromatography and analysed with triple quadrupole mass spectrometry. Complete method description and instrument parameters, including thorough validation of the analytical method according to EMA guidelines (European Medicines Agency 2011), are reported separately (Nandania *et al*, 2018). In blood, 94 metabolites were measured. However, at the time when we performed the muscle metabolite analysis, our metabolite set was updated to 111, including methionine intermediates and acylcarnitines (Dataset EV2).

**Statistical analysis**

Targeted metabolomics data were analysed using MetaboAnalyst 3.0 (www.metabolanalyst.ca; Xia *et al*, 2009, 2015). The data were log-transformed and autoscaled before statistical analysis. Plasma metabolomes of MIRAS ($n = 9$), PEO ($n = 6$), MELAS ($n = 5$) and MIRAS carriers ($n = 16$) were compared to plasma of controls ($n = 30$). Serum metabolomes of IOSCA ($n = 5$), IBM ($n = 5$) and NMD ($n = 15$) patients were compared to serum of controls ($n = 10$). Individual metabolite values are shown for the one additional IOSCA child patient (Fig 3B), to show the relevance of IOSCA findings in early- vs late-stage disease. However, this child patient was not included in the overall statistical analysis of adult IOSCA patients due to lack of appropriate age- and gender-matched control samples. Muscle metabolomes of MIRAS ($n = 5$) and PEO ($n = 5$) patients were compared to muscle of controls ($n = 10$ and $n = 7$, respectively). Differences between control and patient groups were tested with univariate analysis, two-sample t-test. Metabolites were tested for false positivity (FDR) with Benjamini–Hochberg method with a critical value of 0.2 (Dataset EV2). For multivariate regression, we performed partial least squares–discriminant analysis (PLS-DA) with variable importance in projection (VIP). The cross-validation of PLS-DA model was done with leave-one-out cross-validation (LOOCV) method (Table EV2; MetaboAnalyst 3.0). Due to the small amount of female MIRAS and PEO patients, we tested the effect of gender on blood metabolome among our controls (females $n = 16$, males $n = 14$). Three metabolites were significantly changed between male and female controls (Fig EV1A); however, their FDR was >0.7. Therefore, we included all male and female controls in MIRAS and PEO blood analysis (all figures). Due to small amount of MELAS muscle samples ($n = 2$), statistical analysis was not possible (Fig 4). Global test was used for the pathway

enrichment analysis, and relative-betweenness centrality method was used for pathway topology analysis (MetaboAnalyst 3.0; Dataset EV3). Sensitivity and specificity were analysed by the univariate ROC analysis, and AUC was determined (GraphPad PRISM 6; GraphPad Software, La Jolla, CA). A mean centroid for metabolites with the highest AUC (cystathionine, alanine, sorbitol and myoinositol) was calculated for each patient as an overall predictive value (Dataset EV2) and tested with one-way ANOVA and Dunnett's multiple comparison test (GraphPad PRISM 6). The mean centroid values of the four-metabolite biomarker of controls, IOSCA, MIRAS, PEO and MELAS, were used for sensitivity and specificity determination by ROC curve, and AUC was calculated (GraphPad PRISM 6). Serum FGF21 was tested with Kruskal–Wallis test with Dunn's multiple comparisons test (GraphPad PRISM 6). Creatine/creatinine ratio between controls and patients was tested with Mann–Whitney test (GraphPad PRISM 6; Dataset EV2).

## Data availability

The datasets produced in this study are available in the following databases:

Metabolomics data: PeptideAtlas accession number: PASS01255 (http://www.peptideatlas.org/PASS/PASS01255).

**Expanded View** for this article is available online.

### Acknowledgments

The authors wish to thank all the patients and their relatives for participation in the study. Markus Innilä is thanked for patient sampling and technical assistance, and Jatin Nandania is thanked for metabolomics sample analyses. Jenni Lehtonen and Christopher Jackson are thanked for valuable discussion and critical reading of the manuscript. This work was supported by European Research Council, Academy of Finland (Centre of Excellence in Research on Mitochondria, Metabolism and Disease (FinMIT), grant number: 272376), Sigrid Juselius Foundation, Jane and Aatos Erkko Foundation, University of Helsinki and Helsinki University Central Hospital (A.S., K.H.P.), Doctoral Programme in Integrative Life Science (J.B.), Academy of Finland postdoctoral fellowship (C.J.C), Gyllenberg Foundation, Novo Nordisk Foundation, The Finnish Diabetes Research Foundation (K.H.P.), Biocenter Finland (V.V).

### Author contributions

JB conceived experimental design and performance, analysed data and wrote the manuscript. JN and SA analysed data and participated in manuscript writing. AHH recruited and examined patients, and participated in manuscript writing. TH analysed data. KS, HY-J, KHP and TL recruited and examined patients, and edited the manuscript. VV supervised metabolomic experiments and analysed data, and participated in manuscript writing. CJC designed and supervised the study, participated in manuscript writing. AS designed and supervised the study, recruited patients, analysed and interpreted data, and wrote the manuscript.

### Conflict of interest

AS and JB have filed a patent application of the multi-biomarker. No other potential conflict of interests relevant for this article were reported.

### For more information

(i)   International MitoPatients: https://www.mitopatients.org
(ii)  United Mitochondrial Disease Foundation: http://www.umdf.org
(iii) Suomalainen Wartiovaara Lab/University of Helsinki: https://www.helsinki.fi/en/researchgroups/mitochondrial-medicine
(iv)  FIMM Metabolomics/University of Helsinki: https://www.fimm.fi/en/services/technology-centre/metabolomics

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
