## [Review Process File · EMBO Molecular Medicine]

Metabolomes of mitochondrial diseases and inclusion body myositis patients: treatment targets and biomarkers

Jana Buzkova, Joni Nikkanen, Sofia Ahola, Anna H. Hakonen, Ksenia Sevastianova, Topi Hovinen, Hannele Yki-Järvinen, Kirsi H. Pietiläinen, Tuula Lönnqvist, Vidya Velagapudi, Christopher J. Carroll, Anu Suomalainen

Review timeline:

Submission date:	08 March 2018
Editorial Decision:	19 April 2018
Decision Appealed:	20 April 2018
Editorial Decision:	20 April 2018
Revision received:	02 July 2018
Editorial Decision:	24 July 2018
Revision received:	27 August 2018
Editorial Decision:	30 August 2018
Revision received:	14 September 2018
Accepted:	24 September 2018

Editor: Céline Carret

Transaction Report:

1st Editorial Decision

19 April 2018

Thank you for the submission of your manuscript "Metabolomes of mitochondrial diseases and inclusion body myositis: treatment targets and biomarkers". We have now heard back from the three referees whom we asked to evaluate your manuscript.

Unfortunately, you will see that while the referees do find the overall goal interesting and clinically important (as we did), they're not convinced by the methodology and therefore do not find the conclusions sufficiently supported by the data. Of particular relevance, ref 2 (an expert in metabolomics) and ref 3 have issues about the method as reflected in their cross-commenting comments, please see below:

ref2: "There is no question that the topic being addressed by the authors is an area of unmet medical needs, and the findings, if trustworthy, would have been of interest, in agreement with Reviewer 1. The sample size is an inherent issue in this line of research, given the low prevalence of the disorders, and special attention must be paid to the way data is analyzed and interpreted. While setting up an independent cohort might be too challenging at the present time, I agree with the Reviewer 3 that this would have been essential before any firm conclusions can be made about the clinical significance of the findings. Nevertheless, even within the current study setting, proper statistical analyses would have allowed at least to provide robust estimates (with CIs) of the between-group discrimination. Instead, the authors appear to have performed a rather superficial data analysis, which very likely overstated the significance of the findings. Additionally, in the present manuscript, there are potentially serious yet unstated weaknesses in analytical methodology..."

ref3: "The results strongly reflect the methods and without replication it is difficult to be confident in any conclusions."

Given these evaluations, I am sorry to say that I cannot offer further processing at this stage.

I hope, however, that the referees' comments will be helpful to improve the paper.

***** Reviewer's comments *****

Referee #1 (Remarks for Author):

This is interesting work based on the following question: what are the general metabolic consequences of mitochondrial disease and is it possible to identify metabolic signatures that will a) help us to understand what is happening and b) make useful biomarkers. The work focuses on blood, but includes data from muscle. The authors show that there are indeed signatures/fingerprints that help to distinguish specific forms of mitochondrial disease and that inclusion body myositis appears to have a similar profile, thus lending support to the view that mitochondrial dysfunction plays a role in this disorder.

I think the data is sound and the authors' interpretation appropriate. There are always problems interpreting large data sets, particularly with multiple analyses, but this in no way invalidates their data. The authors have attempted to control for specific tissue signatures in muscle by including non-mitochondrial muscle diseases, and a similar analysis for those with purely CNS involvement would be interesting, albeit very difficult to do. A more significant confounder of the metabolic data is the presence of diabetes: while it is not clearly stated, at least 2 of those with the m.3243A>G had DM. I would like to know if the authors controlled for this in any way?

I am also interested in the PEO group: patients with single and multiple deletions are grouped together, yet single deletions might be expected to give a mitochondrial translational defect similar to those with MTTL1 mutation. Is diabetes thus the difference between these?

Comments to the authors

1. The authors picked 94 metabolites in their blood analyses and 110 in their analysis of muscle; I may have missed the explanation, but what were these choices based on?
2. Age at investigation and age of onset differ markedly in the patient groups. Interestingly, in the IOSCA group, patients had an age at sampling ranging from 33 - 42. This is interesting since disease onset is infantile, and raises the question of whether these patients were intrinsically different. Age is also a confounder in the other end of the spectrum: controls had a median age of 42 (blood) and 48.5 (muscle), while we are only given the spread of age for the IBM patients. The NMD ranged from 2 years up to 60.
3. On a similar theme, what are the metabolic consequences of inactivity? This is particularly relevant for the studies of muscle, especially the turnover of creatine/creatinine, and relevant to those with muscle disease including IBM.
4. I found the results concerning arginine very interesting and wonder whether the authors would comment further? There is no confirmed evidence that acute episodes in patients with m.3243A>G are due to vasoconstriction and no real proof that they respond to arginine, even though physicians in some countries have now adopted this treatment as standard. If arginine does have any effect, however, it might be more relevant to think that this was secondary to supplementation rather than any effect on vascular contractility.

Referee #2 (Remarks for Author):

The authors address an important clinical question of specific biomarkers for distinct mitochondrial disorders (MDs), which could be used for monitoring disease progression and treatment efficacy. Since mitochondrial diseases have a major impact on metabolism, as already demonstrated by the authors and others in earlier experimental studies, metabolomics is a promising platform for the discovery of such markers. Sample size is an inherent challenge and limitation in such biomarker discovery studies due to low disease prevalence, thus one cannot expect to reach sample numbers as typically used in biomarker studies in more common diseases.

In the reported study, serum metabolomes were investigated in five MDs, non-mitochondrial neuromuscular diseases and matched controls. The authors claim to have found distinct signature for each MD, and that a panel of four metabolites differentiated between the primary MDs with good sensitivity and high specificity. Also, the underlying pathways were identified.

While the reported results are very promising, there are some potential weaknesses in the underlying methods which are however central to the reported results and their interpretation.

The metabolomics method description is scarce. When following-up the trail of references that the article was referring to for method description, this led to the paper Nikkanen J et al, *Cell Metab* 2016, where also the method was insufficiently described, but in the supplement of that paper another paper was referred to, by Roman-Garcia et al, *J Clin Invest* 2014, which however contained the same level of detail as in Nikkanen J et al. Generally, the trail of method references when using this method leads to the above-mentioned paper or to Khan NA et al, *EMBO Mol Med* 2014 - which does not provide more detail either.

A total of 94 metabolites were measured, while in one of the supplements of Nikkanen et al., 116 metabolites were listed. It is thus unclear which metabolites were measured in the present paper. Since the method is non-standard, claiming to be (1) quantitative while using only a single extraction method, (2) covering a diverse range of metabolites which by basic principles of chemistry cannot all be extracted optimally as needed for the quantitative method, and (3) the metabolites reported are found over a range of concentrations that exceeds the linear range of the mass spectrometer - this surely should be considered as a non-standard method that would need to be thoroughly evaluated by expert analytical chemist (i.e. by publishing it in specialized analytical journal) and presented in sufficient detail. Typically, quantitative analyses require optimized extraction protocols for specific classes of chemically similar metabolites, and thus require multiple analytical runs.

The trail of references that the authors cite for the metabolomics method does not give any data on the method performance, what are the standard compounds, what are the internal standards, whether the method is validated in terms of robustness, linear range, recovery etc. It only gives the UPLC-QqQMS conditions and also them only partially. For example, the MS parameters are not described in sufficient detail, e.g. MRM parameters are not stated at all (this is a requirement for any MS/MS analysis). As the method is non-standard, not a well-established method used in the scientific community, these parameters should be given. Particularly, HILIC methods (applied here) have been demonstrated to be less robust than RPLC, both in terms of variability of retention times (more complex and yet poorly understood retention mechanism in HILIC) as well as in terms of impact of matrix effects.

The rationale for the choice of extraction protocol is poorly explained. Despite being mainly targeted to very polar metabolites and claiming to be quantitative, there is very high volume of organic solvent and thus these metabolites may not be efficiently extracted with the method even in principle. No data are shown on the recovery of the extraction. If the method is stated to be quantitative, the recoveries should be given.

In addition, there is no data on any type of quality control during the analytical analyses, e.g. the use of pooled samples for quality control, use of certified reference material (available for plasma from NIST, with published reference values for at least part of the analytes here) etc.

Since the authors claim that the method is quantitative, this would mean that labelled internal standards were used for each of the 94 metabolite and calibration curves for each metabolite. Is that correct? Quite often, the targeted methods in metabolomics are using only limited number of labelled ISTDs, and then the method is not strictly speaking quantitative due to the unavoidable matrix effects in the LC-MS method. Particularly, it is well known that (underivatized) amino acids suffer from the matrix effects when using HILIC, and that their analysis under acidic conditions in HILIC is particularly problematic, and thus, their quantitation would definitely require individual ISTDs. Here, several amino acids were reported significant, therefore it would be crucial to show data on the method performance. Several published methods using HILIC state that indole derivatives are poorly retained with HILIC; yet here the authors have reported two of these types of compounds. Is it possible to measure them reliably with the method?

The concentration range of the listed analytes may also be problematic. For example, the typical concentration of several compounds are in low nanomolar range (e.g. neopterin, carnosine, glycocholic acid, kynurenate) while several of the compounds also listed are present in high micromolar range. The instrument used for the analysis does not have a linear dynamic range that would allow reliable quantitative analysis over this range of concentrations in a single analysis.

Taken together, from the analytical perspective there is insufficient detail provided to be able to evaluate the reported results with confidence.

Metabolic pathway analysis as applied is heavily dependent on the metabolic coverage. It should be stated how many metabolites in the stated pathway could be detected. While the method is easy to use, the results should be treated with more caution. When simply copy-pasting the list of 116 metabolites from Nikkanen et al 2016 into MetaboAnalyst, this reviewer found almost the same list of pathways as reported here. Some of the reported pathways are quite extensive, e.g. about 50 in bile acid pathways and 60 in cysteine/methionine metabolism, but the reported method is unlikely able to cover more than a couple metabolites from each. Therefore, more likely than not, the reported pathway analysis results are simply the consequence of the number of metabolites covered in each pathway.

Leave-one-out cross-validation was used for PLS-DA analysis to discriminate between the groups. This approach however is prone to over-fitting. Despite the small sample sizes, more proper cross-validation procedures such as random subsets may still be feasible, leading to more conservative and robust results. The method performance incl R2 and Q2 values should also be reported.

It is also unclear how the diagnostic performance of the 4-metabolite panel was evaluated. What was the underlying statistical model and how were the confidence intervals estimated? If the approach in PLS-DA analysis gives any clue, the reported AUCs are likely to be over-optimistic.

In summary, this is a potentially important study which shows that metabolomics may hold promise in the efforts to identify clinically useful markers for MDs. However, analytical methodology is poorly described and may have considerable yet unstated limitations, while the statistical methods are unsound for the purpose.

Referee #3 (Remarks for Author):

The authors have carried out a hypothesis-free metabolomic analysis of a heterogeneous group of mitochondrial, neurological, neuromuscular diseases and controls. They construct a model of four metabolites distinguishing mitochondrial disorders from the other groups. The key issue here is whether the sample sizes are adequate, given the heterogeneity of the groups, and the thousands of possible outcomes in a study of this kind. In my view, there needs to be independent validation before any firm conclusions can be reached. It is also unwise to conclude that the 'metabolic fingerprints' will be of any use in disease follow-up, because no longitudinal data is presented.

Decision Appealed

20 April 2018

Thank you for the feedback. I find the feedback of Ref #1 and #2 constructive and good, and both of them see the value of the analysis. Most of the lengthy criticism is about lacking details of methodology, which we acknowledge, and can provide. However, concerning the comments of Rev #3, it is not realistic that we would collect a parallel second material - from another centre - from such rare disorders. This replication would be done by others to follow. The human samples are rare, and maybe surprisingly, the value of serum materials in mitochondrial disease field is only starting to be recognized and therefore serum collections should be done anew. Importantly, we are replicating in humans findings previously reported in mitochondrial disease mouse models, which we could emphasize more.

So, I would like to ask your opinion: if we can satisfactorily respond to the first two reviewers, would you still reconsider?

2nd Editorial Decision

20 April 2018

Thank you for your letter asking us to reconsider our decision on your article referenced above.

After careful examination of the referees' comments again and intense discussion within the office, we feel that we can consider a major revision of your study, provided that you can make a strong case and convince referee 2 of the methodology chosen. Seemingly, all major concerns raised by the referees should be convincingly addressed (except indeed providing an independent cohort or increasing n).

The technical details requested by referee 2 are absolutely needed. Should you find yourself in a position where you cannot provide all of the data, or agree with limitations (cf linear dynamic range of the instrument), please make sure to thoroughly discuss the issues. I would encourage you to also try to assess statistics as recommended (random subsets rather than leave-one-out cross validation).

I look forward to seeing a revised form of your manuscript within three months.

Please note that EMBO Molecular Medicine strongly supports a single round of revision and that, as acceptance or rejection of the manuscript will depend on another round of review, your responses should be as complete as possible.

I look forward to receiving your revised manuscript.

2nd Revision - authors' response

02 July 2018

Authors' responses to the Reviewers

We would like to thank the Reviewers for their constructive comments. Please find our detailed response below

Referee #1:

“This is interesting work based on the following question: what are the general metabolic consequences of mitochondrial disease and is it possible to identify metabolic signatures that will a) help us to understand what is happening and b) make useful biomarkers. The work focusses on blood, but includes data from muscle. The authors show that there are indeed signatures/fingerprints that help to distinguish specific forms of mitochondrial disease and that inclusion body myositis appears to have a similar profile, thus lending support to the view that mitochondrial dysfunction plays a role in this disorder. I think the data is sound and the authors' interpretation appropriate. There are always problems interpreting large data sets, particularly with multiple analyses, but this in no way invalidates their data. The authors have attempted to control for specific tissue signatures in muscle by including non-mitochondrial muscle diseases, and a similar analysis for those with purely CNS involvement would be interesting, albeit very difficult to do.”

We would like to thank the Reviewer for the highly positive comments.

Comment:

“A more significant confounder of the metabolic data is the presence of diabetes: while it is not clearly stated, at least 2 of those with the m.3243A>G had DM. I would like to know if the authors controlled for this in any way?”

Response:

Thank you for your comment. We agree with the Reviewer that DM can modify metabolism. Decreased insulin sensitivity is a feature of some, but not all MELAS patients, and also of some PEO and MIRAS patients. Because of the Reviewer's comment, we analysed the data now so that we selected all the diabetic mitochondrial disease patients and compared them against the non-diabetics. Although six metabolites were significantly changed between the groups (see below), these were not the same that defined mitochondrial diseases from controls. Also, the false-discovery rates were high, and all exceeded our FDR-limit of 0.2, suggesting that mitochondrial disease was a stronger contributor than diabetes in generating the metabolic fingerprints. We now included these results also in the manuscript (p. 7; Table EV1). We unfortunately do not have available control samples from patients with diabetes, because our controls were selected to represent healthy population.

Metabolites	FC	Raw p-value	FDR
AMP	2.5	0.008	0.35096
Glyceraldehyde	-1.4	0.009	0.35096
Ribose-5-P	2.4	0.0144	0.37573
Glutathione reduced	-2.4	0.0216	0.42107
Phenylalanine	-1.5	0.0412	0.55405
Tryptophan	-1.2	0.0426	0.55405

Comment:

"I am also interested in the PEO group: patients with single and multiple deletions are grouped together, yet single deletions might be expected to give a mitochondrial translational defect similar to those with MTTL1 mutation. Is diabetes thus the difference between these? "

Response:

Despite the molecular background, the clinical manifestation of our PEO patients with single mtDNA deletions is indistinguishable from that of PEO patients with multiple mtDNA deletions (Twinkle-PEO), pointing to similar pathophysiological changes in their muscle and therefore similar clinical outcome. These patients with pure mitochondrial myopathy are clinically quite different to MELAS/MIDD patients (MTTL1 mutation), the latter having often also hearing loss, cardiomyopathy, diabetes and strokes. We therefore grouped the patients based on their carefully studied phenotype, also considering but not restricted by the genotype. This strategy is supported by our previous pilot study with "modified Atkins diet" (Ahola et al. 2016). In this study we also grouped Twinkle-PEO and single-mtDNA-deletion PEO patients, and they responded in a remarkably uniform manner and timeline to the diet, regardless of the genotype, suggesting that the phenotype of manifestation reflects the physiological changes, probably even more than the actual molecular genetic defect. The basis for patient grouping has now been commented in the methods of the manuscript, p. 18.

Of diabetes: our single-mtDNA-deletion PEO patients do not have DM, two Twinkle-PEO patients have mild impairment of glucose tolerance (but do not use medication). Please see the comment above.

Comment:

Comments to the authors

"1. The authors picked 94 metabolites in their blood analyses and 110 in their analysis of muscle; I may have missed the explanation, but what were these choices based on? "

Response:

We apologize for the lack of clarity in the description of the analysed metabolites; the reason is related to the time when the samples were analysed. These rare patient samples were collected during many years, and when the early sets were analysed fresh (as the metabolites do not store well over many years even if frozen), the metabolomic set of ours included 94 metabolites. However, during the years when more samples were collected and analysed, the metabolite analysis had been updated with methionine intermediates and acylcarnitines (total n of metabolites = 111; by mistake one metabolite was missing from the original manuscript; this was now updated in the revised manuscript). We could not sample the patients again to analyse all with the larger set. However, the results from the tissue analysis supported the blood findings. We have now explained the number of analysed metabolites in the manuscript text (p.19). **All the metabolites measured in blood and muscle, as well as the percentage of detected metabolites in each group/ tissue are listed in Dataset EV2.**

Comment:

“2. Age at investigation and age of onset differ markedly in the patient groups. Interestingly, in the IOSCA group, patients had an age at sampling ranging from 33 - 42. This is interesting since disease onset is infantile, and raises the question of whether these patients were intrinsically different. Age is also a confounder in the other end of the spectrum: controls had a median age of 42 (blood) and 48.5 (muscle), while we are only given the spread of age for the IBM patients. The NMD ranged from 2 years up to 60.”

Response:

We thank the Reviewer for the interesting comment concerning IOSCA patients. We have recently diagnosed a young IOSCA patient (4 years old), whose metabolome was now analysed and included for comparison (Fig 2B). Interestingly, already at the early disease stage, especially the levels of creatine and taurine metabolites cluster with the adult-patients, strengthening the specificity of the metabolite signatures for IOSCA disease and pointing to insufficient methyl-pool. We chose to show the individual metabolite values of the young IOSCA patient but because of the lack of appropriate age- and gender-matched control samples for this child, we did not include her in the statistical analyses (p. 20). Of the age-ranges of the controls: we always had two age- and gender matched controls for each patient, and therefore the controls are fully representative. The age-ranges for the controls have now been specified in the text (p. 19).

Comment:

“3. On a similar theme, what are the metabolic consequences of inactivity? This is particularly relevant for the studies of muscle, especially the turnover of creatine/creatinine, and relevant to those with muscle disease including IBM.”

Response:

Indeed, the muscle inactivity could contribute to the creatine/ creatinine ratio in blood. We found increased ratio in IOSCA, IBM and NMD patients, who have various degrees of movement difficulty, but this is true also for MIRAS and to some degree to MELAS, which both clustered separate from the controls. Moreover, the IOSCA child patient (mentioned above) is still motorically as active as her age-mates, but has quite a high creatine/creatinine ratio in blood, similar to adult IOSCA patients (Fig 2B). Therefore, especially in IOSCA we find creatine metabolism interesting and relevant to the disease pathogenesis. Furthermore, knowledge of creatine deficiency, previously unknown to exist in the disease groups, including IBM, might motivate creatine supplementation for these patients to improve muscle function, as has been reported in muscle dystrophy (as discussed in the manuscript). We have now included a comment of inactivity and creatine in the discussion, p. 14.

Comment:

“4. I found the results concerning arginine very interesting and wonder whether the authors would comment further? There is no confirmed evidence that acute episodes in patients with m.3243A>G are due to vasoconstriction and no real proof that they respond to arginine, even though physicians in some countries have now adopted this treatment as standard. If arginine does have any effect, however, it might be more relevant to think that this was secondary to supplementation rather than any effect on vascular contractility. “

Response:

We agree with the Reviewer that the finding of low L-arginine specifically in MELAS in our unbiased metabolite screening was interesting, because of the original Japanese reports of low blood arginine and beneficial effects of L-arginine treatment in MELAS (Koga et al. 2005). Actually, recent data from US and Canada do also support the clinical benefit of L-arginine supplementation (Koenig et al. 2016) in MELAS, despite the fact that the molecular mechanism is still open. Our human data do not allow speculation of the mechanistic details, unfortunately, but motivate further studies on L-arginine in MELAS.

Referee #2:

Comment:

“The metabolomics method description is scarce. When following-up the trail of references that the article was referring to for method description, this led to the paper Nikkanen J et al, Cell Metab 2016, where also the method was insufficiently described, but in the supplement of that paper another paper was referred to, by Roman-Garcia et al, J Clin Invest 2014, which however contained the same level of detail as in Nikkanen J et al. Generally, the trail of method references when using this method leads to the above-mentioned paper or to Khan NA et al, EMBO Mol Med 2014 - which does not provide more detail either. “

Response:

We thank the Reviewer for careful reading. However, all the instrument parameters, reagents, running conditions and the sample handling were already published in detail in Khan et al. 2014 EMBO Mol Med (as supplementary information). The only part that was missing from Khan et al. were the exact MRMs of the measured metabolites, which was published in the Supplementary Table 1 of Kolho et al. 2017. These two references have been now added to the manuscript (p.19). Despite that fact that the details have been reported in previous manuscripts of Khan et al. and Kolho et al, we agree that a separate method validation paper would be helpful. To fully meet this and other comments of this Reviewer, **we have now written such a method article, specifying carefully all the analytical details.** The manuscript has been submitted, and is currently available in BioRxives (Nandania et al. 2018) and also provided here for the Reviewer, attached.

Comment:

A total of 94 metabolites were measured, while in one of the supplements of Nikkanen et al., 116 metabolites were listed. It is thus unclear which metabolites were measured in the present paper.

Response:

We apologize for the lack of clarity. These rare patient samples were collected during many years. When the early sets were analysed fresh, the metabolomic set of ours included 94 metabolites. However, during the years when more samples were collected and analysed, the metabolite analysis had been updated with methionine intermediates and acylcarnitines (total n of metabolites = 111; by mistake one metabolite was missing from the original manuscript; this was now updated in the revised manuscript). We could not collect new samples from the early patients to allow second analysis with a larger set. However, the results from the muscle and blood analysis supported each other. We have now explained the number of analysed metabolites in the manuscript text (p.19). **All**

the metabolites measured in blood and muscle, as well as the percentage of detected metabolites in each group/ tissue are listed in Dataset EV2.

Comment:

Since the method is non-standard, claiming to be (1) quantitative while using only a single extraction method, (2) covering a diverse range of metabolites which by basic principles of chemistry cannot all be extracted optimally as needed for the quantitative method, and (3) the metabolites reported are found over a range of concentrations that exceeds the linear range of the mass spectrometer - this surely should be considered as a non-standard method that would need to be thoroughly evaluated by expert analytical chemist (i.e. by publishing it in specialized analytical journal) and presented in sufficient detail. Typically, quantitative analyses require optimized extraction protocols for specific classes of chemically similar metabolites, and thus require multiple analytical runs.

Response:

During every analysis, we run all the pure compounds as standards to produce the external calibration curves (11 points) for each and every compound, and also use 12 labelled internal standards (listed in our Validation manuscript Table S1, attached).

Linearity was checked for all the metabolites during the validation. Metabolite's concentrations outside the linearity range were evaluated with extended linearity range until the detector's saturation. All the details can be found in our Validation manuscript, attached.

However, to respond to the concern of the Reviewer and to be conservative in our statements, we have now revised the wording to “semi-quantitative”.

Comment:

“The trail of references that the authors cite for the metabolomics method does not give any data on the method performance, what are the standard compounds, what are the internal standards, whether the method is validated in terms of robustness, linear range, recovery etc. It only gives the UPLC-QqQMS conditions and also them only partially. For example, the MS parameters are not described in sufficient detail, e.g. MRM parameters are not stated at all (this is a requirement for any MS/MS analysis). As the method is non-standard, not a well-established method used in the scientific community, these parameters should be given. Particularly, HILIC methods (applied here) have been demonstrated to be less robust than RPLC, both in terms of variability of retention times (more complex and yet poorly understood retention mechanism in HILIC) as well as in terms of impact of matrix effects.”

Response:

Our method has been validated in terms of robustness, linearity, accuracy, precision, selectivity, specificity, recovery, matrix effect, and stability. All the parameters were evaluated according to the European Medicines Agency (EMA) guidelines for bioanalytical method validation. **Please see our separate validation manuscript, available in BioRxives and also provided for the reviewer, attached.**

As specified above, all the details are in Supplementary Data of Khan et al. 2014 and Kolho et al. 2017. Unfortunately the reference of the MRM parameters (Kolho et al. 2017; p.19) had been left out from the current manuscript, but now is included.

HILIC method was also evaluated for retention time variability, where retention time reproducibility was checked for 25 batches over the period of one year and %CV was found below 10% for all the metabolites (separate validation manuscript, attached).

Comment:

“The rationale for the choice of extraction protocol is poorly explained. Despite being mainly targeted to very polar metabolites and claiming to be quantitative, there is very high volume of

organic solvent and thus these metabolites may not be efficiently extracted with the method even in principle. No data are shown on the recovery of the extraction. If the method is stated to be quantitative, the recoveries should be given.”

Response:

We have now revised the wording to more conservative, “semi-quantitative”. See explanation above.

Please see the manuscript of validation, attached. We have selected protein precipitation method as this method is non-selective and hence, extraction of metabolites with diverse polarity would be possible. We have selected Acetonitrile as an organic solvent as it is moderate non-polar solvent and hence, most of the metabolites are soluble in this solvent. Also, recovery and matrix factor were evaluated during the method validation.

Comment:

“In addition, there is no data on any type of quality control during the analytical analyses, e.g. the use of pooled samples for quality control, use of certified reference material (available for plasma from NIST, with published reference values for at least part of the analytes here) etc.”

Response:

Again, please see the manuscript of validation for more details and quality assurance, attached. We have a strict quality management system and always use internal quality control (QC) samples (pooled healthy human serum) during each and every analytical batch (every 5th run is a blank and every 10th run is a QC). We have used QC samples to check the reliability of our method. Variation in QC samples data was always investigated after completion of every batch. We have also analysed the certified standard reference material, plasma from NIST (SRM 1950), using our method and compared the 17 matched metabolite’s concentrations. The correlation co-efficient between the standard reference values and the values obtained from our method is $r^2 = 0.967$. We have also done cross-platform comparison, where we have sent our internal QC samples to the NMR facility, Kuopio, Finland; and also, to the BIOCRATES Life Sciences AG, Innsbruck, Austria to check the robustness and performance of our method using completely different extraction protocols and analytical platforms. We are able to match 38 metabolites from the BIOCRATES Absolute IDQ p180 metabolomics assay kit, and the correlation co-efficient between the BIOCRATES values and the values obtained from our method is $r^2 = 0.975$. We are able to match 22 metabolites from the NMR small molecules analysis, and the correlation co-efficient between the NMR values and the values obtained from our method is $r^2 = 0.884$. Please see the manuscript of validation for more details and quality assurance, attached.

Thus, we can confidently state that our method is a “standard and well-established method”, where reputed international and national institutes and researches have been widely using (Szibor et al. 2017; Schatton et al. 2017; Ali-Sisto et al. 2018; Rey et al. 2016; Scott et al. 2017; Roman-Garcia et al. 2014).

Comment:

“Since the authors claim that the method is quantitative, this would mean that labelled internal standards were used for each of the 94 metabolite and calibration curves for each metabolite. Is that correct? Quite often, the targeted methods in metabolomics are using only limited number of labelled ISTDs, and then the method is not strictly speaking quantitative due to the unavoidable matrix effects in the LC-MS method. Particularly, it is well known that (underivatized) amino acids suffer from the matrix effects when using HILIC, and that their analysis under acidic conditions in HILIC is particularly problematic, and thus, their quantitation would definitely require individual ISTDs. Here, several amino acids were reported significant, therefore it would be crucial to show data on the method performance. Several published methods using HILIC state that indole derivatives are poorly retained with HILIC; yet here the authors have reported two of these types of

compounds. Is it possible to measure them reliably with the method? The concentration range of the listed analytes may also be problematic. For example, the typical concentration of several compounds are in low nanomolar range (e.g. neopterin, carnosine, glycocholic acid, kynurenate) while several of the compounds also listed are present in high micromolar range. The instrument used for the analysis does not have a linear dynamic range that would allow reliable quantitative analysis over this range of concentrations in a single analysis.”

Response:

The reviewer is absolutely correct regarding the common practice in targeted high-throughput metabolomics analyses, where people use only limited number of labelled internal standards. We have used 12 labelled internal standards for amino acids and for other compounds depending up on their chemical classes. Please see the manuscript of validation, attached. Individual external calibration curves were used, where r^2 variation is mostly <3%, and also, there is <4% variability in the retention time. Linearity was checked for all the metabolites during the validation. Metabolite's concentrations outside the linearity range were evaluated with extended linearity range until the detector's saturation. In addition, for all the metabolites, matrix effect and process efficiency were evaluated to check ion suppression at RT of metabolites including indole derivatives. Furthermore, metabolites without internal standards were always corrected with process efficiency. We have now, however, changed the term to describe the method as “semi-quantitative”. According to the Human Metabolome Database, HMDB (www.hmdb.ca), the listed metabolites (neopterin, carnosine, glycocholic acid, kynurenate) have concentrations in micromolar range which are well in agreement with our reported values (Fonteh et al. 2007; Spiller et al. 1987; Durantont et al. 2012).

Comment:

“Taken together, from the analytical perspective there is insufficient detail provided to be able to evaluate the reported results with confidence. “

Response:

We wish that we have now responded in fine detail to all the concerns of this reviewer.

Comment:

“Metabolic pathway analysis as applied is heavily dependent on the metabolic coverage. It should be stated how many metabolites in the stated pathway could be detected. While the method is easy to use, the results should be treated with more caution. When simply copy-pasting the list of 116 metabolites from Nikkanen et al 2016 into MetaboAnalyst, this reviewer found almost the same list of pathways as reported here. Some of the reported pathways are quite extensive, e.g. about 50 in bile acid pathways and 60 in cysteine/methionine metabolism, but the reported method is unlikely able to cover more than a couple metabolites from each. Therefore, more likely than not, the reported pathway analysis results are simply the consequence of the number of metabolites covered in each pathway. “

Response:

One of the main points of this paper is that we actually replicate in human patients the metabolic findings characterized previously in mitochondrial disease mice that were reported by Nikkanen et al. 2016, Khan et al. EMBO Mol Med 2014 and Cell Metab 2017. In these articles, we extensively validated the metabolomic findings with independent methods, including testing of one-carbon cycle pathway outputs by protein, enzyme activity and RNA analysis, even in vivo metabolic flux with untargeted metabolomics approach. This kind of replication of metabolomic data with independent methods can be done in disease models, but not in human patients. We are happy to see that our original findings are now also replicated by others in the field, studying various models of mitochondrial dysfunction, and also extensively validated with independent methods (for example,

Bao et al. 2016; Kühl et al. 2017). To our view there is no question of whether one-carbon cycle outputs are affected in these models or not. In human blood samples we cannot unfortunately target the pathways as we can with mice. The strength of this paper is that it replicates the mouse data and emphasizes the relevance and conservation of the mouse findings in human patients. The method and metabolite setup has also been previously used in many other disorders as well (for example inflammatory bowel disease in Kolho et al. 2017), not lighting up these pathways.

To respond to this comment, we have now revised the figure 4 to show the number of hits in every pathway.

We selected for the Figure 4 only pathways with >10% of hits/ pathway and included the number of hits/ total number of metabolites in the pathway to the figure picture (Fig 4). **The full list of pathways for each diseases group, including the detected metabolites in each pathway, is available in Dataset EV3.**

Comment:

“Leave-one-out cross-validation was used for PLS-DA analysis to discriminate between the groups. This approach however is prone to over-fitting. Despite the small sample sizes, more proper cross-validation procedures such as random subsets may still be feasible, leading to more conservative and robust results. The method performance incl R2 and Q2 values should also be reported. “

Response:

We have utilized a widely accepted and standard analysis (Metaboanalyst) for the analysis of the metabolomics data, which also was used to originally identify the metabolic changes in the mouse models (Nikkanen et al. 2016; Khan et al. 2017), rigorously validated in independent experiments. **To meet the comments of the Reviewer, we have now log-transformed the data and included R2 and Q2 values to Table EV2.** The Q2 predictive values for all disease groups are acceptable, except for MIRAS muscle samples that shows very low Q2 values. However, this result is not surprising, since the MIRAS patients have mostly a CNS phenotype and only mild or no muscle signs, and therefore the muscle metabolomics profile of these patients is similar to controls.

Comment:

“It is also unclear how the diagnostic performance of the 4-metabolite panel was evaluated. What was the underlying statistical model and how were the confidence intervals estimated? If the approach in PLS-DA analysis gives any clue, the reported AUCs are likely to be over-optimistic. “

Response:

We have now clarified the method description, as requested by the Reviewer. For the 4-metabolite biomarkers, the mean centroid values of primary mitochondrial disease patients (IOSCA, MIRAS, PEO and MELAS) were used to determine the sensitivity and specificity by ROC curve, and the AUC was calculated. We included this information also in the statistical methods of the manuscript (p.20 - 21).

In mitochondrial diseases that are rare, the kind of material that we have is actually quite representative. Additionally, we were able to collect two more MELAS patients and one IOSCA child patient that were now included. These samples strengthened the original conclusions. We have also tuned down the biomarker part somewhat, as that was a concern for the reviewer. We did not, however, want to omit it because it is potentially quite important.

Comment:

“In summary, this is a potentially important study which shows that metabolomics may hold promise in the efforts to identify clinically useful markers for MDs. However, analytical methodology is poorly described and may have considerable yet unstated limitations, while the statistical methods are unsound for the purpose.”

Response:

We are happy that the Reviewer sees the value of our data. We wish to have now responded to all the concerns of the Reviewer.

Referee #3Comment:

“The authors have carried out a hypothesis-free metabolomic analysis of a heterogeneous group of mitochondrial, neurological, neuromuscular diseases and controls. They construct a model of four metabolites distinguishing mitochondrial disorders from the other groups. The key issue here is whether the sample sizes are adequate, given the heterogeneity of the groups, and the thousands of possible outcomes in a study of this kind. In my view, there needs to be independent validation before any firm conclusions can be reached. It is also unwise to conclude that the 'metabolic fingerprints' will be of any use in disease follow-up, because no longitudinal data is presented. “

Response:

We would like to point out that MIRAS and IOSCA patients carry ancestral homozygous mutations and are of Finnish origin; the population shows less variation than populations typically. The clinical phenotypes were similar within the study groups. However, we agree with the Reviewer that independent replication of the data in other cohorts is needed in future. In our opinion, our paper sets the stage in human patients, and is heavily supported by previous mechanistic data from mouse and cell models, by us and others (Nikkanen et al. 2016; Bao et al. 2016; Kühl et al. 2017; Ost et al. 2015), and will be followed up by independent studies in other cohorts. To replicate the findings in another sample set from another centre with MELAS, MIRAS, PEO, IOSCA, IBM, NMD and a number of matched controls is simply not feasible or realistic for this article.

We'd like to emphasize that our careful study includes samples from a high number of age- and gender-matched controls, mitochondrial patients and their family members with genetically verified diagnoses, non-mitochondrial muscle disease controls, and even secondary mitochondrial disease patients (IBM). Such materials are rare and valuable – so far, no such cohorts have been published. Indeed, an important point that was raised during the study was the similarity of the IBM metabolome with that of muscle-manifesting mitochondrial diseases. The data have high relevance for understanding the role of mitochondrial dysfunction in IBM symptoms and also for treatment strategies of this relatively common disease. We were able to increase the MELAS patient number, and also of a rare IOSCA child, now included in the analysis.

References for responses:

- Ahola, Sofia, Mari Auranen, Pirjo Isohanni, Satu Niemisalo, Niina Urho, Jana Buzkova, Vidya Velagapudi, et al. 2016. “Modified Atkins Diet Induces Subacute Selective Ragged-red-fiber Lysis in Mitochondrial Myopathy Patients.” *EMBO Molecular Medicine* 8 (11): 1234–47. doi:10.15252/emmm.201606592.
- Ali-Sisto, Toni, Tommi Tolmunen, Heimo Viinamäki, Pekka Mäntyselkä, Minna Valkonen-Korhonen, Heli Koivumaa-Honkanen, Kirsi Honkalampi, et al. 2018. “Global Arginine Bioavailability Ratio Is Decreased in Patients with Major Depressive Disorder.” *Journal of Affective Disorders* 229 (March). Elsevier: 145–51. doi:10.1016/j.jad.2017.12.030.
- Bao, Xiaoyan Robert, Shao-En Ong, Olga Goldberger, Jun Peng, Rohit Sharma, Dawn A Thompson, Scott B Vafai, et al. 2016. “Mitochondrial Dysfunction Remodels One-Carbon Metabolism in Human Cells.” *ELife* 5. doi:10.7554/eLife.10575.
- Duranton, F., G. Cohen, R. De Smet, M. Rodriguez, J. Jankowski, R. Vanholder, and A. Argiles. 2012. “Normal and Pathologic Concentrations of Uremic Toxins.” *Journal of the American Society of Nephrology* 23 (7): 1258–70. doi:10.1681/ASN.2011121175.
- Fonteh, A. N., R. J. Harrington, A. Tsai, P. Liao, and M. G. Harrington. 2007. “Free Amino Acid and Dipeptide Changes in the Body Fluids from Alzheimer's Disease Subjects.” *Amino Acids* 32 (2): 213–24. doi:10.1007/s00726-006-0409-8.
- Khan, Nahid A., Mari Auranen, Ilse Paetau, Eija Pirinen, Liliya Euro, Saara Forsström, Lotta

- Pasila, et al. 2014. “Effective Treatment of Mitochondrial Myopathy by Nicotinamide Riboside, a Vitamin B3.” *EMBO Molecular Medicine* 6 (6): 721–31. doi:10.1002/emmm.201403943.
- Khan, Nahid A., Joni Nikkanen, Shuichi Yatsuga, Christopher Jackson, Liya Wang, Swagat Pradhan, Riikka Kivelä, Alberto Pessia, Vidya Velagapudi, and Anu Suomalainen. 2017. “MTORC1 Regulates Mitochondrial Integrated Stress Response and Mitochondrial Myopathy Progression.” *Cell Metabolism* 26 (2): 419–428.e5. doi:10.1016/j.cmet.2017.07.007.
- Koenig, Mary Kay, Lisa Emrick, Amel Karaa, Mark Korson, Fernando Scaglia, Sumit Parikh, and Amy Goldstein. 2016. “Recommendations for the Management of Strokelike Episodes in Patients With Mitochondrial Encephalomyopathy, Lactic Acidosis, and Strokelike Episodes.” *JAMA Neurology* 73 (5). American Medical Association: 591–94. doi:10.1001/jamaneurol.2015.5072.
- Koga, Y, Y Akita, J Nishioka, S Yatsuga, N Povalko, Y Tanabe, S Fujimoto, and Toyojiro Matsuishi. 2005. “L-Arginine Improves the Symptoms of Strokelike Episodes in MELAS.” *Neurology* 64 (4). Wolters Kluwer Health, Inc. on behalf of the American Academy of Neurology: 710–12. doi:10.1212/01.WNL.0000151976.60624.01.
- Kolho, Kaija-Leena, Alberto Pessia, Tytti Jaakkola, Willem M De Vos, and Vidya Velagapudi. 2017. “Faecal and Serum Metabolomics in Paediatric Inflammatory Bowel Disease.” *Journal of Crohn's and Colitis*, 321–34. doi:10.1093/ecco-jcc/jjw158.
- Kühl, Inge, Maria Miranda, Ilian Atanassov, Irina Kuznetsova, Yvonne Hinze, Arnaud Mourier, Aleksandra Filipovska, and Nils-Göran Larsson. 2017. “Transcriptomic and Proteomic Landscape of Mitochondrial Dysfunction Reveals Secondary Coenzyme Q Deficiency in Mammals.” *ELife* 6 (November). doi:10.7554/eLife.30952.
- Nandania, Jatin, Gopal Peddinti, Alberto Pessia, Meri Kokkonen, and Vidya Velagapudi. 2018. “Validation and Automation of a High-Throughput Multi-Targeted Method for Semi-Quantification of Endogenous Metabolites from Different Biological Matrices Using Tandem Mass Spectrometry.” *BioRxiv*, June. Cold Spring Harbor Laboratory, 352468. doi:10.1101/352468.
- Nikkanen, J., S. Forsström, L. Euro, I. Paetau, R.A. Kohnz, L. Wang, D. Chilov, et al. 2016. “Mitochondrial DNA Replication Defects Disturb Cellular DNTP Pools and Remodel One-Carbon Metabolism.” *Cell Metabolism* 23 (4). doi:10.1016/j.cmet.2016.01.019.
- Ost, Mario, Susanne Keipert, Evert M. van Schothorst, Verena Donner, Inge van der Stelt, Anna P. Kipp, Klaus-Jürgen Petzke, et al. 2015. “Muscle Mitohormesis Promotes Cellular Survival via Serine/Glycine Pathway Flux.” *FASEB Journal : Official Publication of the Federation of American Societies for Experimental Biology* 29 (4): 1314–28. doi:10.1096/fj.14-261503.
- Rey, Guillaume, Utham K. Valekunja, Kevin A. Feeney, Lisa Wulund, Nikolay B. Milev, Alessandra Stangherlin, Laura Ansel-Bollepalli, Vidya Velagapudi, John S. O’Neill, and Akhilesh B. Reddy. 2016. “The Pentose Phosphate Pathway Regulates the Circadian Clock.” *Cell Metabolism* 24 (3): 462–73. doi:10.1016/j.cmet.2016.07.024.
- Roman-Garcia, Pablo, Isabel Quiros-Gonzalez, Lynda Mottram, Liesbet Lieben, Kunal Sharan, Arporn Wangwiwatsin, Jose Tubio, et al. 2014. “Vitamin B12–dependent Taurine Synthesis Regulates Growth and Bone Mass.” *Journal of Clinical Investigation* 124 (7): 2988–3002. doi:10.1172/JCI72606.
- Schatton, Désirée, David Pla-Martin, Marie Charlotte Marx, Henriette Hansen, Arnaud Mourier, Ivan Nemazanyy, Alberto Pessia, et al. 2017. “CLUH Regulates Mitochondrial Metabolism by Controlling Translation and Decay of Target MRNAs.” *The Journal of Cell Biology*. doi:10.1083/jcb.201607019.
- Scott, Timothy A., Leonor M. Quintaneiro, Povilas Norvaisas, Prudence P. Lui, Matthew P. Wilson, Kit-Yi Leung, Lucia Herrera-Dominguez, et al. 2017. “Host-Microbe Co-Metabolism Dictates Cancer Drug Efficacy in *C. Elegans*.” *Cell* 169 (3): 442–456.e18. doi:10.1016/j.cell.2017.03.040.
- Spiller, R C, P F Frost, J S Stewart, S R Bloom, and D B Silk. 1987. “Delayed Postprandial Plasma Bile Acid Response in Coeliac Patients with Slow Mouth-Caecum Transit.” *Clin Sci (Lond)* 72 (2): 217–23. doi:10.1042/cs0720217.
- Szibor, Marten, Praveen K Dhandapani, Eric Dufour, Kira M Holmström, Yuan Zhuang, Isabelle Salwig, Ilka Wittig, et al. 2017. “Broad AOX Expression in a Genetically Tractable Mouse Model Does Not Disturb Normal Physiology.” *Disease Models & Mechanisms* 10 (2). The Company of Biologists Ltd: 163–71. doi:10.1242/dmm.027839.

Thank you for the submission of your manuscript to EMBO Molecular Medicine. We have now heard back from the two referees whom we asked to re-evaluate your manuscript.

You will see that while referee 1 is fully satisfied, still highlighting that s/he is not a metabolomics person, referee 2 does not support publication.

While ref2 's report is rather clear, we wondered whether there could be a way for you to satisfy the referee without having to re-run the samples. Unfortunately, this is a case where the technology and analyses are so important and critical to the main message of the paper that if the expert in this technology finds some flows, the conclusions are weakened. This is why we have asked referee 2 to recommend further analyses to make the paper better suited for publication.

Please find below ref.2's responses to my prompting:

"Indeed, I thought that given [that] this is presented as a kind of 'diagnostic' paper, presenting biomarker candidates in rare samples, it is particularly important that methodological issues are properly addressed.

[about the cross-validation approach not being adequate] Concerning PLS-DA, this is a potentially addressable issue. I have already stated previously that they should try with resampling or other more conservative CV approach which is usually adopted in PLS-DA. LOO is rarely used, precisely because of overfitting; I suspect the authors use that because it gave them still reasonably good results given the relatively small group sizes; they should at least show how it works with more conservative and robust methods, and then discuss limitations if it doesn't quite work out.

[about linearity and validation] Analytical methods are rarely perfect, but at the very least they should be transparently presented. I think the authors should be very clear about what they report and what the limitations are. It does appear the method description was quickly assembled as it does not appear solid, that's why I stated a lot of work remains to be done on the method (for the performance as described in the paper to be true). Showing 'linearity' data on log-log scale is really strange, it does appear that the authors are hiding something. This is not a good practice in metabolomics/analytical chemistry and scientifically completely unacceptable.

[about having to re-run all samples] However, I don't think they would need to rerun the samples if the method is clearly described and limitations stated, some improvements could be done with the already acquired data. Specifically, given the calibration looks problematic (although appears 'linear' due to log transformation), they could still use 'semi-quantitative' normalization by using metabolite peak areas divided by internal standard peak areas. "

I hope that you will be willing to address referee 2's concerns as thoroughly a possible. Please note that EMBO Molecular Medicine usually allows only a single round of revision and therefore, this is indeed the last chance. Your revised paper will have to be evaluated by referee 2 once more.

I look forward to seeing a revised form of your manuscript as soon as possible.

***** Reviewer's comments *****

Referee #1 (Remarks for Author):

The authors have re-submitted the manuscript that now includes 2 new mitochondrial patients including a child with IOSCA (cf: my comment on age related influence). My comments concerning the aims/questions are the same and I think that the dataset is improved and remains sound. I accept that that the controls were chosen as "normal" and thus a comparison with known diabetics is not possible. It is however interesting to see that diabetes does generate a profile and that, based on their data/interpretation, it appears that mitochondrial dysfunction is a stronger influence than diabetes for the metabolic changes. This will need to be validated, but does not detract from the findings pertaining to mitochondrial disease.

The authors have addressed my concerns regarding the influence of age and inactivity and explained the choice of metabolites.

I found the work both interesting and important. I feel that the additional data, re-interpretation and revision have improved it further. I am not a metabolomics person, and have, therefore, not commented on the technical aspects of this analysis. I think, however, that the manuscript contains valuable insights and that the interpretation of the data is robust. My decision remains, therefore, that it is worthy of publication.

Referee #2 (Remarks for Author):

The authors clarified their analytical approach by correcting the wording describing the method, as well as submitted an analytical paper describing the method, which is commendable. However, there appear to be several issues with the analytical approach, and most likely, there is much work to be done (see specific comments on the analytical method below).

Regarding bioinformatics, the authors partly addressed the issues raised by providing additional detail about their approach. Particularly, the pathway analysis is now clearly described and presented, although the limitations, which remain, are not critically discussed.

However, the cross-validation (LOO) approach still used in PLS-DA is prone to over-fitting. Given the inherent limitation of the study of having small number of patients per study group, it would be essential to apply a more conservative approach. Referring to MetaboAnalyst as a 'standard analysis' is not a good argument to justify the approach, particularly given the small sample sizes.

As a general comment, it is disappointing that the authors do not critically discuss the limitations of the study in the discussion.

Comments on the analytical method

The authors have done a lot of work for the method validation. However, there are several major issues in the presented approach.

'Linearity'

The authors have been transforming both the concentrations and the responses using either logarithmic or square root functions. This will make even the most nonlinear response look linear, which is why such data transformation, when showing linearity of the response in quantitation, is not an accepted practice in analytical chemistry. Even when the concentrations are far above the detectors actual linear range (i.e. the detector is saturated) the calibration curve will look linear with this approach. A calibration curve on linear scale as low as $R^2 < 0.6$ (e.g. saturated response) will get, after log transformation, a curve that does look linear ($R^2 > 0.85$). This approach also makes the repeatability look much better than it actually is, making even huge actual differences much smaller (e.g. with a variation of the actual response $> 50\%$ can be diminished to ca. 10%).

Validation

It is not possible to study the selectivity and specificity in the manner than the authors have been doing. They have used a protocol typically used for targeted analysis in other applications (e.g. pharmaceutical analyses), where a blank sample is available. However, in metabolomics, there is no blank matrix available for most metabolites. Furthermore, they have used only one type of matrix, i.e. serum samples in this part. Even the different serum samples can differ substantially (even in the same type of species) and definitely in different types of biological samples. Also, the chromatogram showing this experiment shows very clearly that the amount spiked was very high, so the concentrations were most probably not representing any biological levels. Also, high concentration spikes will minimize the effects of possible matrix interferences or effects of any co-eluting matrix compounds, thus making the results look good while not giving a realistic view.

Recoveries and matrix effects cannot be calculated in a way the authors have done it. Again, this approach is suitable only when blank matrices (which are not available for metabolomics) are available, especially for the matrix effects, see the discussion for specificity. As this is not possible in metabolomics, the matrix effects should be studied by using a different approach. e.g. by direct infusion of each compound separately while injecting sample (each matrix). The recoveries for biopsy samples particularly cannot be estimated in this way, as spiked compounds are not bound to the matrix, so they are easily extracted, unlike the compounds in the actual sample matrix.

Sample carry-over should be measured with real samples, not with standards, as even the name of the experiments suggests - sample carry-over. Standard samples are much cleaner, while the real samples can cause interferences also due to co-eluting matrix components, not only the metabolites of interest. Thus, using standard compounds only will give far too optimistic picture. Again, different sample types may also have a very different carry-over.

Additional communication with Referee #2

Indeed, I thought that given this is presented as a kind of 'diagnostic' paper, presenting biomarker candidates in rare samples, it is particularly important that methodological issues are properly addressed.

Concerning PLS-DA, this is a potentially addressable issue. I have already stated previously that they should try with resampling or other more conservative CV approach which is usually adopted in PLS-DA. LOO is rarely used, precisely because of overfitting; I suspect the authors use that because it gave them still reasonably good results given the relatively small group sizes; they should at least show how it works with more conservative and robust methods, and then discuss limitations if it doesn't quite work out.

Analytical methods are rarely perfect, but at the very least they should be transparently presented. I think the authors should be very clear about what they report and what the limitations are. It does appear the method description was quickly assembled as it does not appear solid, that's why I stated a lot of work remains to be done on the method (for the performance as described in the paper to be true). Showing 'linearity' data on log-log scale is really strange, it does appear that the authors are hiding something. This is not a good practice in metabolomics/analytical chemistry and scientifically completely unacceptable.

However, I don't think they would need to rerun the samples if the method is clearly described and limitations stated, some improvements could be done with the already acquired data. Specifically, given the calibration looks problematic (although appears 'linear' due to log transformation), they could still use 'semi-quantitative' normalization by using metabolite peak areas divided by internal standard peak areas.

3rd Revision - authors' response

27 August 2018

Revision (round 2)

We do appreciate that the Reviewer 2 showed detailed interest on reviewing our manuscript. We are however, quite puzzled about the continuously critical comments with few suggestions, claiming our method to be non-standard and not a well-established method used in the scientific community, despite of it being used in numerous established peer-reviewed articles (30 examples listed in the end of these responses, i.e. over 60 reviewers before this one have accepted the methodology to be standard and well-established: Refs 1-30). We are also puzzled why our previous revisions and additional analyses were not much commented (for example inclusion of NIST SRM plasma reference standards; cross-platform comparisons; a separate methods paper), even if performed and explained exactly as requested.

For the information for the Reviewer, our separate validation article of the method (provided in the 1st revision) has **now been reviewed by three expert reviewers from the fields of metabolomics/analytical chemistry, and accepted after minor revisions for publication in the**

journal “Metabolites” (PMID:30081599), an official journal of the International Metabolomics Society, USA. Our method was considered appropriate, state-of-the-art, and reliable. **Further, the analysis methods strictly follow the guidelines of European Medicine Agency (EMA)**, as detailed below. **These aspects should make it absolutely clear that the method and the results are solid and robust.** Furthermore, **these first-of-its-kind human patient data beautifully replicate** pathological pathways originally established and validated with independent proteomics, metabolomics and protein biochemical methods in different model systems by us and many other high-profile research groups (listed below). The data importantly indicate the pathways to be relevant both for patients with rare primary and common secondary mitochondrial disorders.

Referee #2 (Remarks for Author): The authors clarified their analytical approach by correcting the wording describing the method, as well as submitted an analytical paper describing the method, which is commendable.

Response: Thank you for acknowledging our separate validation article. We did not only “correct wording”, but analyzed and provided further standards for the analysis, exactly as requested by the reviewer, as detailed below.

Regarding bioinformatics, the authors partly addressed the issues raised by providing additional detail about their approach. Particularly, the pathway analysis is now clearly described and presented, although the limitations, which remain, are not critically discussed.

Response: We have now included further discussion on the limitation of the discussion p.16, of the small sample size & issues of PLS-DA and overfitting.

However, the cross-validation (LOO) approach still used in PLS-DA is prone to over-fitting. Given the inherent limitation of the study of having small number of patients per study group, it would be essential to apply a more conservative approach. Referring to MetaboAnalyst as a 'standard analysis' is not a good argument to justify the approach, particularly given the small sample sizes. As a general comment, it is disappointing that the authors do not critically discuss the limitations of the study in the discussion.

Response: We agree with the Reviewer that no consensus of the perfect statistical analysis of metabolomic data exists in the literature. We would like to point that our data are not standing alone, but similar pathways have been found in several independent studies on different mouse and cell models by us and others, but not in patient materials before [Nikkanen et al. Cell Metab 2016 (metabolomics plus protein biochemical validation in mouse models with mitochondrial dysfunction); Bao et al., 2016 ELife (metabolomics and proteomics of cell lines with challenged mitochondrial dysfunction); Köhl et al., 2017 ELife (several different mouse models with mitochondrial disease, proteomics, metabolomics)].

In our study, multivariate PLS-DA separated the patient groups well from matched controls, and gave overlapping data in individual analysis of muscle-manifesting mitochondrial disease patient groups. The finding that muscle-manifesting mitochondrial diseases share serum metabolite profiles in mice (Nikkanen et al. Cell Metab 2016, ref. 27) has been shown before, and also found by our current study – importantly also applying to secondary, common, sporadic muscle disorders with secondary mitochondrial dysfunction.

We had already in the previous revision included in the Supplementary data the R2 and Q2 values of the PLS-DA model [upper and lower limit of the model's ability to predict observations; provided by MetaboAnalyst, calculated together with the cross validation]. In PLS-DA, we minimised overfitting by choosing a low-dimensional model, without excessive analysis components, as proposed by Xia & Wishart, Curr Protoc Bioinformatics (2016). We also utilized univariate (t-test with FDR correction) in addition to the multivariate (PLS-DA) data analysis, a standard method (see list of 30 references in the end). Our biomarker analysis was performed with univariate ROC analysis, not PLS-DA.

We wish that our conservative and critical analysis and state-of-the-art methodology, filling all strict criteria of state-of-the-art metabolomics analysis, is now acceptable for the Reviewer. To further

address the Reviewer's concern, we included a study limitation paragraph in the discussion now, p.16.

Linearity: The authors have been transforming both the concentrations and the responses using either logarithmic or square root functions. This will make even the most nonlinear response look linear, which is why such data transformation, when showing linearity of the response in quantitation, is not an accepted practice in analytical chemistry.

Response: We are puzzled about this comment, because in high-throughput metabolomics analysis, where usually bioanalytical methods cover a wide dynamic concentration range, log-log transformation is the typical approach for presentation of heteroscedastic data with broad concentration range (i.e., variance increases as the concentration increases; for example see Dempo et al., 2014; Dubbelman et al., 2018). We evaluated several regression models and best fitted each calibration curve with appropriate weighing factors and transformations. There are numerous publications supporting this approach; for example:

For example: Singtoroj, T.; Tarning, J.; Annerberg, A.; Ashton, M.; Bergqvist, Y.; White, N.J.; Lindegardh, N.; Day, N.P. *A new approach to evaluate regression models during validation of bioanalytical assays. J Pharm Biomed Anal.* **2006**, *41*, 219-217, DOI: 10.1016/j.jpba.2005.11.006. "2.1.4. Transformation methods: An alternative approach to overcome heteroscedastic data is to transform x and/or y before constructing the regression line. Two common approaches are logarithmic or square root transformation of both x and y before Ordinary Linear Regression".

In the above article, the authors had **evaluated 19 different regression models with over 1000-fold concentration range** and their conclusion is, "*The results showed that log-log transformation without weighting was the simplest model to fit the calibration data and ensure good predictability for this data set*". This is in line with our approach too.

Even when the concentrations are far above the detectors actual linear range (i.e. the detector is saturated) the calibration curve will look linear with this approach.

Response: As already mentioned in our previous comments to the reviewer, the linearity was checked for each and every metabolite separately in this semi-quantitative method (targeted metabolomics), so there is no question of touching the detector's saturation limit. In addition, metabolite concentrations outside the linearity range were evaluated with extended linearity range until the detector's saturation.

A calibration curve on linear scale as low as $R^2 < 0.6$ (e.g. saturated response) will get, after log transformation, a curve that does look linear ($R^2 > 0.85$). This approach also makes the repeatability look much better than it actually is, making even huge actual differences much smaller (e.g. with a variation of the actual response $> 50\%$ can be diminished to ca. 10%). Showing 'linearity' data on log-log scale is really strange, it does appear that the authors are hiding something.

Response: The detector's response does not need to be saturated while building a calibration curve. In our calibration mix, each calibration curve has over 1000-fold wide dynamic concentration range. We statistically evaluated several regression models and best fitted each calibration curve with appropriate weighing factors and transformations. Log-log transformation is a common practice to cover a broad concentration range and recommended approach for our type of analysis as explained above.

This is not a good practice in metabolomics/analytical chemistry and scientifically completely unacceptable.

Response: We are not aware of any guidelines or references, making the point that log-log transformation in broad calibration curves in metabolomics is scientifically completely unacceptable? The available literature suggests the opposite – it is the recommended approach. – as pointed out by the references above. Therefore we decided to follow the consensus and not modify these results.

However, I don't think they would need to rerun the samples if the method is clearly described

Response: We are pleased to hear that the Reviewer accepts the current analyses. In our separate article about the method, now accepted for publication in a methodological journal “Metabolites”¹ (PMID:30081599) the method has been thoroughly described, as well as in the 30 references listed below.

... and limitations stated, some improvements could be done with the already acquired data. Specifically, given the calibration looks problematic (although appears 'linear' due to log transformation), they could still use 'semi-quantitative' normalization by using metabolite peak areas divided by internal standard peak areas. "

Response: As stated in the manuscript, our approach **is** semi-quantitative, targeted metabolomics, not untargeted, as probably mistaken by the reviewer. Our analyses follow recommendations in the field of semi-quantitative analysis approaches, as detailed above.

Validation: It is not possible to study the selectivity and specificity in the manner than the authors have been doing. They have used a protocol typically used for targeted analysis in other applications (e.g. pharmaceutical analyses), where a blank sample is available. However, in metabolomics, there is no blank matrix available for most metabolites. Furthermore, they have used only one type of matrix, i.e. serum samples in this part. Even the different serum samples can differ substantially (even in the same type of species) and definitely in different types of biological samples.

Response: First: we would like to make the point that we have done our validation exactly according to EMA guidelines for bioanalytical methods.
http://www.ema.europa.eu/docs/en_GB/document_library/Scientific_guideline/2011/08/WC500109686.pdf

Secondly, all biological or clinical samples have inter-individual and sample-type variability. Therefore, naturally, in our study we have sample type-, age- and gender-matched controls for patient serum and plasma samples, and matched control muscle for patient muscle samples. The samples have been collected after overnight fasting, and this approach is as close to standardization as one can get when analyzing human samples. As serum samples (and plasma, and muscle etc) differ from person to person, we then ask whether we can find significant changes with the material available, and report the significantly changed variables. **This is completely standard and state-of-the-art in the field of human disease research.** If this is not accepted, then human studies in general are not acceptable, and all studies should be done with rodents or lower model organisms, which of course have inter-individual variation also.

To encounter the analysis variation, we prepare fresh 11-point calibration curves in every analytical run and check for any interferences from metabolites at their retention times in the chromatography, and each analyte has a unique MRM transition that we evaluate in the mass spec analysis, which reduces the selectivity problem. Most importantly our original results (Nikkanen et al. Cell Metab 2016, ref. 27) have been robustly replicated in independent studies in different mouse and cell models (e.g. Bao et al. ELife 2016 (Mootha-group); Kuhl et al. ELife 2017 (NG Larsson group)). **However, the current report is the first one in human mitochondrial patients, importantly indicating that one-carbon cycle imbalance is an important manifestation of mitochondrial dysfunction, with disease-specific features, biomarker potential and targets for treatment.**

To the best of our knowledge, we have not found any articles from the metabolomics community, where selectivity and specificity are validated using another approach.

Also, the chromatogram showing this experiment shows very clearly that the amount spiked was very high, so the concentrations were most probably not representing any biological levels. Also, high concentration spikes will minimize the effects of possible matrix interferences or effects of any co-eluting matrix compounds, thus making the results look good while not giving a realistic view.

Response: We show in our validation article in “Metabolites” (now the Reviewer is commenting the methods article, not the manuscript under review currently) a representative diagram, and the chromatograms for all the metabolites in our QC serum sample are available as a supplementary file in our validation article¹.

Recoveries and matrix effects cannot be calculated in a way the authors have done it. Again, this approach is suitable only when blank matrices (which are not available for metabolomics) are available, especially for the matrix effects, see the discussion for specificity. As this is not possible in metabolomics, the matrix effects should be studied by using a different approach. e.g. by direct infusion of each compound separately while injecting sample (each matrix).

Response: As mentioned, we followed carefully the EMA guidelines and our method's validation paper was recently reviewed and accepted for publication in the "Metabolites" journal. Numerous previous articles from other groups also followed EMA guidelines in the metabolomics analysis for matrix effects. As an example, the following reference is given for the Reviewer's information.

"Matrix effect-corrected liquid chromatography/tandem mass-spectrometric method for determining acylcarnitines in human urine" by Kazuki Abe et al Clinica Chimica Acta (2017). <https://www.sciencedirect.com/science/article/pii/S000989811730075X>

There are no strict guidelines on which matrix effect methodologies are to be performed. Reviewer's suggestion of "direct infusion" method is a qualitative approach that identifies the chromatographic regions where analyte would be susceptible to ion suppression or enhancement, whereas the method we followed is a quantitative approach. In addition to being qualitative, direct infusion approach is quite time consuming and requires significant optimization for each and every compound, which is not a desirable approach for high-throughput methods, where hundreds of analytes are measured in a single assay like ours. The Reviewer's suggestion might be a method of choice for pure analytical chemistry, for quantitative analysis of a handful of similar compounds, so that they can evaluate each compound and adjust the chromatography accordingly. To the best of our knowledge, no metabolomics approach claimed that the direct infusion method is the only correct approach to calculate the matrix effects in metabolomics as there is no blank matrix.

As the reviewer does not provide any alternative guidelines or references, which specified that a) only direct infusion is acceptable in metabolomics analysis to address matrix effects, or b) how the direct infusion approach addressed better the blank matrix issues than the quantitative approach, or c) how direct infusion is done efficiently when analyzing hundreds of metabolites from tens of different classes in just 17.5 min of chromatographic run time, we keep following the accepted EMA guidelines.

We do follow GLP and do regular maintenance of the instrument and optimized sample preparation steps to reduce the matrix effect. It is well known that endogenous phospholipids and proteins are major source of matrix effects. Thus, we use protein precipitation extraction method and we always use specialized Ostro™ 96-well plate (Waters Corporation, Milford, USA), which contain phospholipid chelating agents in each filter. Hence, we obtain a much cleaner extract. Furthermore, we also centrifuge cleaner extracts just before the LC injection.

As mentioned in the validation article "the challenge of the matrix effect can be overcome by having individual isotope-labeled internal standards for each individual compound for true quantification. However, this is not practically possible for high-throughput metabolomics analyses. This is due to high costs and also because not all internal standards are commercially available. In our method, we selected 12 labeled internal standards, which represent chemically similar classes for optimal correction".

Since only 12 isotopically labelled standards cannot compensate the matrix effect for all the 102 compounds, we described our method as "semi-quantification" method. Moreover, we correct the metabolites, that are without internal standards, with the process efficiency factor.

Our excellent QC reproducible and accuracy results for over one year speak strongly against inconsistent matrix effects¹. (QC published in our validation article).

It is a well-known fact that as long as the matrix effects is reproducible it does not necessarily need to be eliminated, but should be identified and quantified. As mentioned in our validation manuscript, *"the repeatability of the matrix effect in terms of CV was less than 25% for most of the compounds. Reliable measurements are accordingly possible"*. We have now added a comment of this aspect also to the current manuscript, to the methods, p.19.

The recoveries for biopsy samples particularly cannot be estimated in this way, as spiked compounds are not bound to the matrix, so they are easily extracted, unlike the compounds in the actual sample matrix.

Response: First of all, as we again described in our validation article¹ (PMID:30081599), “For 85-90 metabolites, recoveries were found to be between 50-120% with good repeatability at all three concentrations levels (low, medium, and high) in both biofluid (serum) and tissues (brain, liver, and spleen)”.

Secondly, we are well aware that it is not possible to determine the exact recovery results using spiking experiment for tissue samples as the spiked standards will not be incorporated into the cells of the tissues. However, the most important aspect to be noted here is that we use exactly the same protocol for all the analysed samples and the repeatability of recoveries at every concentration level (low, medium, and high) in our validation experiment was within CV-15% except for few compounds in some tissues as described in our validation article¹, and now also added to the current manuscript, to the methods, p. 19. Hence the results are comparable within and between the studies.

Because of the reviewer’s open comment without recommendation, we keep using with the state-of-the-art technology.

Sample carry-over should be measured with real samples, not with standards, as even the name of the experiments suggests - sample carry-over. Standard samples are much cleaner, while the real samples can cause interferences also due to co-eluting matrix components, not only the metabolites of interest.

Response: The Reviewer is with this comment now reviewing our currently published methods article, not the current manuscript. There was a typo in the section heading of our validation manuscript and it should be read “Carry-over” instead of “Sample carry-over”. We have performed the carry-over analyses exactly according to EMA guidelines, recommendation being that the carry-over analysis should be done using the highest concentrated pure standards. The EMA-guideline reference has now been added to references, p. 19.

Reference: European Medicines Agency. *Guideline on Bioanalytical Method Validation*; European Medicines Agency: London, UK, 2011.

Thus, using standard compounds only will give far too optimistic picture. Again, different sample types may also have a very different carry-over.

Response: As explained above, we follow exactly the regulatory European EMA guidelines for bioanalytical method validation.

"Indeed, I thought that given [that] this is presented as a kind of 'diagnostic' paper, presenting biomarker candidates in rare samples, it is particularly important that methodological issues are properly addressed".

Response: We cannot agree more with the Reviewer that it is of key importance that methods are validated when novel biomarkers are suggested. We would like to make a few points. 1) Our study is a mechanistic metabolism study, which then identified metabolic signatures partially common for both primary and secondary mitochondrial diseases. This is especially important for IBM, which does not have an established known molecular pathogenesis and the relevance of mitochondrial abnormalities for the disease process are not known. Our data indicates that metabolism in IBM is changed in a similar way as in primary, single-gene mitochondrial diseases, having major implications for understanding IBM pathogenesis and potentially for treatment. 2) as a secondary outcome, we found that specific metabolites were informative as biomarkers, with significant capacity to identify mitochondrial diseases from other muscle-specific diseases. 3) The data robustly replicates data produced in different model systems (by metabolomic and proteomic analyses, by different established research groups in Broad Institute, Max Planck Institute of Aging and in Helsinki University), and therefore this is not a stand-alone metabolomic study.

The methodological concerns of this Reviewer we have now exhaustively addressed above, as well as in our previous response. We wish that these comments are satisfactory for the Reviewer and the Editor.

Analytical methods are rarely perfect, but at the very least they should be transparently presented. I think the authors should be very clear about what they report and what the limitations are.

Response: We completely agree with this comment. It is of utmost importance that the method is well validated, standardized, reproducible and robust. Since the reproducibility still is a major challenge in the metabolomics field, the International Metabolomics Community has been putting a lot of efforts in data standardization, quality control, reproducibility, robustness and data sharing with the goal of moving towards applicability and integration of metabolomics data in Precision/Personalised Medicine. In this context, our method provides a very timely and important contribution to the field. All the methodological aspects have been discussed and are completely transparently presented, and detailed in both the validation article and the current paper.

It does appear the method description was quickly assembled as it does not appear solid (for the performance as described in the paper to be true).

Response: We are puzzled by this critical comment. If the reviewer means the quick progress of the separate validation article for the method (and not the one to be currently reviewed), the manuscript was already drafted before the comments, and recently accepted with minor concerns by three expert reviewers to “Metabolites” journal¹ (PMID:30081599); the rapid acceptance further indicating the solidness of the method per se, with few points of criticism. For this currently reviewed manuscript, we do not see any indication for this comment. In general, we have quality control (QC) data from 6 years; specifically collected reproducibility QC data from 25 different batches from over one year¹; thorough validation results following the European Medicine Agency (EMA) guidelines, and the GLP and quality management is strictly maintained since the foundation of our Metabolomics National Core Facility <https://www.biocenter.fi/index.php/technology-platform-services/proteomics-and-metabolomics#metabol>, with a large international user network and known for its excellent reproducibility.

Although our method is “semi-quantitative”, to address the reviewer’s previous comments, we analysed and compared the NIST SRM plasma reference values against our method and obtained $r^2=0.97$; and also, have done cross-platform comparability with two completely different analytical platforms BIOCRATES kit ($r^2=0.97$) and NMR ($r^2=0.88$). This shows the reliability and robustness of our method. Our method & results fill all the state-of-the-art criteria of the field, and the results are reproducible, and replicate robustly findings previously found in model systems. The data are important, as they report the findings to be relevant for humans, both for primary and secondary mitochondrial diseases.

However, there appear to be several issues with the analytical approach, and most likely, there is much work to be done (see specific comments on the analytical method below). The authors have done a lot of work for the method validation. However, there are several major issues in the presented approach, that's why I stated a lot of work remains to be done on the method.

Response: Please see our responses above. We have exhaustively handled these concerns; to conclude, our analytical method is a standard, well characterized and established, thoroughly validated, reliable, reproducible and robust. Our method has been widely used in the scientific community nationally and internationally, and we published several research articles in the reputed journals. Our validation article has been reviewed by three metabolomics/analytical chemists expert reviewers and accepted for publication in the journal “Metabolites” journal¹ PMID:30081599, an official journal of the International Metabolomics Society, USA, as wished originally by this reviewer too.

References

1. Nandania, J., Peddinti, G., Pessia, A., Kokkonen, M., **Velagapudi, V.^S**. Validation and automation of high-throughput multi-targeted method for semi-quantification of endogenous metabolites from different biological matrices using tandem mass spectrometry. **Metabolites**. 2018; 5:8:3. PMID:30081599

2. Jokinen, R., Rinnankoski-Tuikka, R., Kaye, S., Saarinen, L., Heinonen, S., Myöhänen, M., Rappou, E., Jukarainen, S., Kaprio, J., Rissanen, A., Pessia, A., **Velagapudi, V.**, Virtanen, K., Pirinen, E and Pietiläinen, K. Adipose tissue mitochondrial gene expression profile associates with long-term weight loss success. *International Journal of Obesity*. 2018; 42;817–825. PMID:29203860
3. Zusinaite E, Ianevski A, Niukkanen D, Poranen MM, Bjørås M, Afset JE, Tenson T, **Velagapudi V**, Merits A, Kainov DE. A Systems Approach to Study Immuno- and Neuro-Modulatory Properties of Antiviral Agents. *Viruses*. 2018 Aug 12;10(8). PMID:30103549
4. Muniandy, M., **Velagapudi, V.**, Hakkarainen, A., Lundbom, J., Lundbom, N., Rissanen, A., Kaprio, J., Pietiläinen, K., Ollikainen, M. Plasma metabolites reveal distinct profiles associating with different metabolic risk factors in monozygotic twin pairs. *International Journal of Obesity*. 2018; PMID:29907843
5. De Ruijter, J., Koskela, E., **Nandania, J.**, Frey, A[§], **Velagapudi, V.**[§]. Understanding the metabolic burden of recombinant antibody production in *Saccharomyces cerevisiae* using a quantitative metabolomics approach. *Yeast*. 2018; 35(4):331-341. PMID: 29159981
6. Ali-Sisto, T., Tolmunen, T., Toffol, E., Viinamäki, H., Mäntyselkä, P., Valkonen-Korhonen, M., Honkalampi, K., Ruusunen, A., **Velagapudi, V.**, Lehto, S.M. Global arginine bioavailability ratio is decreased in patients with major depressive disorder. *Journal of Affective Disorders*. 2018; 15;229:145. PMID:29310063
7. Scott TA, Quintaneiro LM, Norvaisas P, Lui PP, Wilson MP, Leung KY, Herrera-Dominguez L, Sudiwala S, **Pessia A**, Clayton PT, Bryson K, **Velagapudi V**, Mills PB, Typas A, Greene NDE, Cabreiro F. (2017). Host-Microbe Co-metabolism Dictates Cancer Drug Efficacy in *C.elegans*. *Cell*. PMID:2843124
8. Bulanova, D., Ianevski, A., Bugai, A., Akimov, Y., Kuivanen, S., Paavilainen, H., Kakkola, L., **Nandania, J.**, Turunen, L., Ohman, T., Ala-Hongisto, H., Pesonen, H.M., Kuisma, M.S., Honkimaa, A., Walton, E.L., Oksenysh, V., Lorey, M.B., Guschin, D., Shim, J., Kim, J., Than, T.T., Chang, S.Y., Hukkanen, V., Kuleskiy, E., Marjomaki, V.S., Julkunen, I., Nyman, T.A., Matikainen, S., Saarela, J.S., Sane, F., Hober, D., Gabriel, G., De Brabander, J.K., Martikainen, M., Windisch, M.P., Min, J.Y., Bruzzone, R., Aittokallio, T., Vähä-Koskela, M., Vapalahti, O., Pulk, A., **Velagapudi, V.**, Kainov, D.E. Antiviral Properties of Chemical Inhibitors of Cellular Anti-Apoptotic Bcl-2 Proteins. *Viruses*. 2017; 9, 271. PMID:28946654
9. Khan, N.A., Nikkanen, J., Yatsuga, S., Wang, L., Jackson, C., **Pessia, A.**, Riikka Kivelä, **Velagapudi, V.**, Anu Suomalainen. mTORC1 Regulates Mitochondrial Folate Cycle and the Integrated Stress Response in Mitochondrial Disease. *Cell Metabolism*. 2017; 26(2):419-428. PMID:28768179
10. Puhka, M., Takatalo, M., Nordberg, M.E., Valkonen, S., **Nandania, J.**, Aatonen, M., Yliperttula, M., Laitinen, S., **Velagapudi, V.**, Mirtti, T., Kallioniemi, O., Rannikko, A., Siljander, P.R.M., af Hallstrom, T.M. Metabolomic profiling of extracellular vesicles and alternative normalisation methods reveal enriched metabolites and strategies to study prostate cancer related changes. *Theranostics*. 2017; 7(16): 3824-3841.
11. Purhonen, J., Rajendran, J., Mörgelin, M., Uusi-Rauva, K., Katayama, S., Krjutskov, K., Einarsdottir, E., **Velagapudi, V.**, Kere, J., Jauhiainen, M., Fellman, V., Kallijärvi, J. (2017). Ketogenic diet attenuates hepatopathy in mouse model of respiratory chain complex III deficiency caused by a Bcs1l mutation. *Scientific Reports*. PMID:28424480
12. Vorrink, S.U., Ullah, S., Schmidt, S., **Nandania, J.**, **Velagapudi, V.**, Beck, O., Ingelman-Sundberg, M., Lauschke, V.M. (2017). Endogenous and xenobiotic metabolic stability of primary human hepatocytes in long-term 3D spheroid cultures revealed by a combination of targeted and untargeted metabolomics. *The FASEB Journal*. PMID:28264975
13. Schatton, D., Pla-Martin, D., Marx, M.C., Hansen, H., Mourier, A., Nemazanyy, I., **Pessia, A.**, Zentis, P., Corona, T. Kondylis, V., Barth, E., Schauss, A.C., **Velagapudi, V.**, Rugarli, E. (2017). CLUH regulates mitochondrial metabolism by controlling translation and decay of target mRNAs. *Journal of Cell Biology*. PMID:28188211
14. Szibor, M., Dhandapani, PK., Dufour, E., Holmström, KM., Zhuang, Y., Salwig, I., Wittig, I., Heidler, J., Gizatullina, Z., Gainutdinov, T., Consortium, GM., Fuchs, H., Gailus-Durner, V., de Angelis, MH., **Nandania, J.**, **Velagapudi, V.**, Wietelmann, A., Rustin, P., Gellerich, FN., Jacobs, HT., Braun, T. (2017). Broad AOX expression in a genetically tractable mouse model does not disturb normal physiology. *Disease Models and Mechanisms*. PMID:28067626
15. Kuivanen, S., Bespalov, MM., **Nandania, J.**, Ianevski, A., **Velagapudi, V.**, De Brabander, JK., Kainov, DE., Vapalahti, O. (2017). Obatoclax, saliphenylhalamide and gemcitabine inhibit Zika

- virus infection in vitro and differentially affect cellular signaling, transcription and metabolism. *Antiviral Research*. PMID:28049006.
16. Gaelings, L., Söderholm, S., Bugai, A., Fu, Y., **Nandania, J.**, Schepens, B., Lorey, M.B., Tynell, J., Ginste, L.S., Goffic, R.L., Miller, M.S., Kuisma, M., Marjomäki, V., Brabander, J.D., Matikainen, S., Nyman, T.A., Bamford, D., Saelens, X., Julkunen, I., Paavilainen, H., Hukkanen, V., **Velagapudi, V.[§]**, Kainov, D.E.[§]. (2017). Regulation of Kynurenine Biosynthesis during Influenza Virus Infection. *The FEBS Journal*. PMID:27860276
 17. Mäntyselkä, P., Ali-Sisto, T., Kautiainen, H., Savolainen, J., Niskanen, L., Viinamäki, H., **Velagapudi, V.**, Lehto, S.M. (2016). The association between musculoskeletal pain and circulating ornithine – A population-based study. *Pain Medicine*. PMID:28034972
 18. Haapaniemi, E., Fogarty, C., Katayama, S., Vihinen, H., Keskitalo, S., Ilander, M., Krjutškov, K., Mustjoki, S., Lehto, M., Hautala, T., Jokitalo, E., **Velagapudi, V.**, Varjosalo, M., Seppänen, M., Kere, J. (2016). Combined immunodeficiency with hypoglycemia caused by mutations in *hypoxia up-regulated 1*. *The Journal of Allergy and Clinical Immunology*. PMID:27913302
 19. Rajendran, J., Tomašić, N., Kotarsky, H., Hansson, E., **Velagapudi, V.**, Kallijärvi, J., Fellman, V. (2016). Effect of high-carbohydrate diet on plasma metabolomics in mice with mitochondrial respiratory chain complex III deficiency due to a *Bes1* mutation. *International Journal of Molecular Sciences*. PMID:27809283
 20. Zinkevičienė A, Kainov D, Girkontaitė I, Lastauskienė E, Kvedarienė V, Fu Y, Anders S, **Velagapudi V**. (2016). Activation of Tryptophan and Phenylalanine Catabolism in the Remission Phase of Allergic Contact Dermatitis: A Pilot Study. *International Archives of Allergy and Immunology*. PMID: 27771694
 21. Kolho, K-L.[§], **Pessia, A.**, Jaakkola, T., de Vos, W., **Velagapudi, V.[§]**. (2016). Fecal and serum metabolomics in pediatric inflammatory bowel disease. *Journal of Crohn's and Colitis*. PMID: 27609529
 22. Ahola, S., Auranen, M., Isohanni, P., Niemisalo, S., Buzkova, J., **Velagapudi, V.**, Lundbom, N., Hakkarainen, A., Piirilä, P., Pietiläinen, K., Suomalainen, A. (2016). Modified Atkins diet induces subacute selective ragged red fiber lysis in mitochondrial myopathy patients. *EMBO Molecular Medicine*. PMID: 27647878 Press release
 23. Rey, G., Valekunja, U.K., Feeney, K.A., Wulund, L., Milev, N.B., Stangherlin, A., Bollepalli, L., **Velagapudi, V.**, O'Neill, J.S., Reddy, A.B. The Pentose Phosphate Pathway Regulates the Circadian Clock. (2016). *Cell Metabolism*. PMID: 27546460
 24. Fu Y, Gaelings L, Söderholm S, Belanov S, **Nandania J**, Nyman TA, Matikainen S, Anders S, **Velagapudi V**, Kainov DE. (2016). JNJ872 inhibits influenza A virus replication without altering cellular antiviral responses. *Antiviral Research*. PMID: 27451344
 25. Schrade, A., Kyrönlahti, A., Akinrinade, O., Pihlajoki, M., Fischer, S., Martinez Rodriguez, V., **Velagapudi V**, Toppari, J., Wilson, DB., Heikinheimo, M. (2016). GATA4 regulates blood-testis barrier function and lactate metabolism in mouse Sertoli cells. *Endocrinology*. PMID: 26974005
 26. Ali-Sisto, T., Tolmunen, T., Toffol, E., Viinamäki, H., Mäntyselkä, P., Valkonen-Korhonen, M., Honkalampi, K., Ruusunen, A., **Velagapudi V**, Lehto, S.M. (2016). Purine metabolism is dysregulated in patients with major depressive disorder. *Psychoneuroendocrinology*. PMID: 27153521
 27. Nikkanen J, Forsström S, Euro L, Paetau I, Kohnz R.A, Wang L, Chilov D, Viinamäki J, Roivainen A, Marjamäki P, Liljenbäck H, Ahola S, Buzkova J, Terzioglu M, Khan N.A, Pirnes-Karhu S, Paetau A, Lönnqvist T, Sajantila A, Isohanni P, Tyynismaa H, Nomura D.K, Battersby B, **Velagapudi V**, Carroll C.J, Suomalainen A. (2016). Mitochondrial DNA Replication Defects Disturb Cellular dNTP Pools And Remodel One-Carbon Metabolism. *Cell Metabolism*. PMID: 26924217
 28. Schrade A, Kyrönlahti A, Akinrinade O, Pihlajoki M, Häkkinen M, Fischer S, Alastalo T.P, **Velagapudi V**, Toppari J, Wilson D.B, Heikinheimo M (2015). GATA4 is a key regulator of steroidogenesis and glycolysis in mouse Leydig cells. *Endocrinology*. PMID: 25668067
 29. Roman-Garcia P, Quiros-Gonzalez I, Mottram L, Lieben L, Sharan K, Wangwiwatsin A, Tubio J, Lewis K, Wilkinson D, Santhanam B, Sarper N, Clare S, Vassiliou GS, **Velagapudi V**, Dougan G, Yadav V.K. (2014). Vitamin B12-dependent taurine synthesis regulates growth and bone mass. *Journal of Clinical Investigation*. PMID: 24911144
 30. Khan N.A, Auranen M, Paetau I, Pirinen E, Euro L, Forsström S, Pasila L, **Velagapudi V**, Carroll C.J, Auwerx J, Suomalainen A. (2014). Effective treatment of mitochondrial myopathy by nicotinamide riboside, a vitamin B3. *EMBO Molecular Medicine*. PMID: 24711540.

31. Bao, X. R., Ong, S.-E., Goldberger, O., Peng, J., Sharma, R., Thompson, D. A., ... Mootha, V. K. (2016). Mitochondrial dysfunction remodels one-carbon metabolism in human cells. *ELife*, 5. <https://doi.org/10.7554/eLife.10575>
32. Dempo, Y., Ohta, E., Nakayama, Y., Bamba, T., & Fukusaki, E. (2014). Molar-Based Targeted Metabolic Profiling of Cyanobacterial Strains with Potential for Biological Production. *Metabolites*. <https://doi.org/10.3390/metabo4020499>
33. Dubbelman, A. C., Cuyckens, F., Dillen, L., Gross, G., Vreeken, R. J., & Hankemeier, T. (2018). Mass spectrometric recommendations for Quan/Qual analysis using liquid-chromatography coupled to quadrupole time-of-flight mass spectrometry. *Analytica Chimica Acta*. <https://doi.org/10.1016/j.aca.2018.02.055>
34. Kühl, I., Miranda, M., Atanassov, I., Kuznetsova, I., Hinze, Y., Mourier, A., ... Larsson, N.-G. (2017). Transcriptomic and proteomic landscape of mitochondrial dysfunction reveals secondary coenzyme Q deficiency in mammals. *ELife*, 6. <https://doi.org/10.7554/eLife.30952>
35. Xia, J., & Wishart, D. S. (2016). Using MetaboAnalyst 3.0 for Comprehensive Metabolomics Data Analysis. *Current Protocols in Bioinformatics*, 55(1), 14.10.1-14.10.91. <https://doi.org/10.1002/cpbi.11>

4th Editorial Decision

30 August 2018

Thank you for the submission of your revised manuscript to EMBO Molecular Medicine and for your response to referee 2' comments. We have looked into the details of that letter, including the now published article describing the methods you have used here. I am happy to say that we decide to move forward with acceptance, pending minor editorial amendments.

Corresponding Author Name: Anu Suomalainen

Manuscript Number: EMM-2018-09091